# TP53 drives invasion through expression of its Δ133p53β variant

Gilles Gadea[1†], Nikola Arsic[2,3†], Kenneth Fernandes[4†], Alexandra Diot[4], Sébastien M Joruiz[4], Samer Abdallah[2,3], Valerie Meuray[4], Stéphanie Vinot[2,3], Christelle Anguille[2,3], Judit Remenyi[4], Marie P Khoury[4], Philip R Quinlan[4], Colin A Purdie[4], Lee B Jordan[4], Frances V Fuller-Pace[4], Marion de Toledo[2,5], Maïlys Cren[2,6], Alastair M Thompson[4,7], Jean-Christophe Bourdon[4*‡], Pierre Roux[2,3*‡]

[1]UM 134 Processus Infectieux en Milieu Insulaire Tropical (PIMIT), INSERM U1187, CNRS UMR9192, IRD UMR249, Université de la Réunion, Sainte Clotilde, France; [2]Université Montpellier, Montpellier, France; [3]CRBM, CNRS, Centre de Recherche de Biologie cellulaire de Montpellier, Montpellier, France; [4]Division of Cancer Research, University of Dundee, Ninewells Hospital and Medical School, Dundee, United Kingdom; [5]CNRS, Institut de Génétique Moléculaire de Montpellier, Montpellier, France; [6]IRB, Institut de Recherche en Biothérapie, Montpellier, France; [7]Department of Surgical Oncology, MD Anderson Cancer Centre, Houston, United States

*For correspondence: j.bourdon@dundee.ac.uk (J-CB); pierre.roux@crbm.cnrs.fr (PR)

†These authors also contributed equally to this work
‡These authors also contributed equally to this work

Competing interests: The author declares that no competing interests exist.

**Abstract** *TP53* is conventionally thought to prevent cancer formation and progression to metastasis, while mutant *TP53* has transforming activities. However, in the clinic, *TP53* mutation status does not accurately predict cancer progression. Here we report, based on clinical analysis corroborated with experimental data, that the p53 isoform Δ133p53β promotes cancer cell invasion, regardless of *TP53* mutation status. Δ133p53β increases risk of cancer recurrence and death in breast cancer patients. Furthermore Δ133p53β is critical to define invasiveness in a panel of breast and colon cell lines, expressing WT or mutant *TP53*. Endogenous mutant Δ133p53β depletion prevents invasiveness without affecting mutant full-length p53 protein expression. Mechanistically WT and mutant Δ133p53β induces EMT. Our findings provide explanations to 2 long-lasting and important clinical conundrums: how WT *TP53* can promote cancer cell invasion and reciprocally why mutant *TP53* gene does not systematically induce cancer progression.

## Introduction

Cancer is driven by somatically acquired point mutations and chromosomal rearrangements, that are thought to accumulate gradually over time. Recent whole cancer genome sequencing studies have conclusively established that the tumor suppressor gene, *TP53,* is the most frequently mutated gene in a wide range of cancer types. In tumors expressing wild-type (WT) *TP53* gene, numerous experimental and clinical data have shown that viruses or cellular oncogene proteins target the p53 pathway, promoting abnormal cell proliferation. Altogether these data strongly suggest that defects in *TP53* tumor suppressor activity are a compulsory step to cancer formation. In addition, ample data have also demonstrated that *TP53* gene, whether WT or mutant, has a paramount biological and clinical role in response to cancer treatment (*Brosh and Rotter, 2009*; *Do et al., 2012*; *Jackson et al., 2012*; *Muller et al., 2009*).

**eLife digest** Most cancers are caused by a build-up of mutations that are acquired throughout life. One gene in particular, called *TP53*, is the most commonly mutated gene in many types of human cancers. This suggests that *TP53* mutations play an important role in cancer development.

It is widely considered that the *TP53* gene normally stops tumors from forming, while mutant forms of the gene somehow promote cancer growth. Evidence from patients with cancer has shown, however, that the relationship between *TP53* mutations and cancer is not that simple. Some very aggressive cancers that resist treatment and spread have a normal *TP53* gene. Some cancers with a mutated gene do not spread and respond well to cancer treatments. Recent studies have shown that the normal *TP53* gene produces many different versions of its protein, and that some of these naturally occurring forms are found more often in tumors that others. However, it was not clear if certain versions of *TP53*'s proteins contributed to the development of cancer.

Now, Gadea, Arsic, Fernandes et al. show that Δ133p53β, one version of the protein produced by the *TP53* gene in human cells, helps tumor cells to spread to other organs. Tests of 273 tumors taken from patients with breast cancer revealed that tumors with the Δ133p53β protein were more likely to spread. Patients with these Δ133p53β-containing tumors were also more likely to develop secondary tumors at other sites in the body and to die within five years.

Next, a series of experiments showed that removing Δ133p53β from breast cancer cells grown in the laboratory made them less likely to invade, while adding it back had the opposite effect. The same thing happened in colon cancer cells grown in the laboratory. The experiments showed that Δ133p53β causes tumor cells with the normal *TP53* gene or a mutated *TP53* gene to spread to other organs.

Together the new findings help explain why some aggressive cancers develop even with a normal version of the tumor-suppressing *TP53* gene. They also help explain why not all cancers with a mutant version of the *TP53* gene go on to spread. Future studies will be needed to determine whether drugs that prevent the production of the Δ133p53β protein can help to treat aggressive cancers.

We previously reported that the human *TP53* gene expresses at least twelve p53 isoforms through alternative splicing of intron-2 (Δ40) and intron-9 (α, β, γ), initiation of transcription in intron-4 (Δ133) and alternative initiation of translation at codon 40 (Δ40) and codon 160 (Δ160). This leads to the expression of p53 (α, β, γ), Δ40p53 (α, β, γ), Δ133p53 (α, β, γ) and Δ160p53 (α, β, γ) protein isoforms that contain different transactivation domain, oligomerisation domains and regulatory domains (for review *Joruiz and Bourdon, 2016*).

All animal models (zebrafish, drosophila and mouse) of p53 isoforms and experimental data in human cells of diverse tissue origins have consistently shown that p53 isoforms regulate cell cycle progression, programmed cell death, replicative senescence, viral replication, cell differentiation and angiogenesis. Several clinical studies reported that abnormal expressions of p53 isoforms are found in a wide range of human cancers including breast and colon cancers and that p53 isoforms are associated with cancer prognosis (*Joruiz and Bourdon, 2016*). However, it is not known whether they are just markers or play an active role in cancer formation and progression. Recently, we reported that Δ133p53β promotes cancer stem cell potential and metastasis formation in a xenograft mouse model (*Arsic et al., 2015*). However, its physiopathological role, its molecular mechanism, its association with cancer progression and the effect of *TP53* mutations on its activities have never been investigated.

To date, it is currently thought that all the tumor suppressor activities associated with *TP53* or the oncogenic activities associated with mutant *TP53* gene expression are implemented by the canonical full-length p53 protein (also named TAp53α or p53α). It is well established that TAp53α , is a transcription factor rapidly activated to restore and maintain cell integrity in response to alteration of cell homeostasis, while most TAp53α missense mutants have compromised tumor suppressor activities and have acquired transforming activities.

Compelling evidence indicates that WT TAp53α protein, plays a pre-eminent role in preventing human cancer formation and progression to metastasis. In particular, WT TAp53α protein prevents cell migration and invasion which are the main traits of Epithelial to Mesenchymal Transition (EMT), which itself involves global genome reprogramming that drives cancer progression to metastasis (*McDonald et al., 2011*; *Suva et al., 2013*). Conversely, most TAp53α missense mutant proteins promote EMT, inducing cancer cells to leave their primary tumor site and disseminate, leading to metastasis (*Bernard et al., 2013*; *Gadea et al., 2007*; *Muller et al., 2009*; *Roger et al., 2010*). However, there remain many enigmas concerning the association of p53 expression with cancer progression and treatment largely due to the natural existence of multiple alternative transcripts from the *TP53* gene. (*De Roock et al., 2009*; *Brosh and Rotter, 2009*; *Petitjean et al., 2007*; *Soussi, 2007*).

The main focus of this study is the role of Δ133p53β in cancer progression. We show that Δ133p53β is associated with poor prognosis in breast cancer, particularly in luminal A breast cancer patients who express WT *TP53.* This indicates that Δ133p53β, which is a physiological gene product of *TP53*, is a predictor of cancer cell invasion and increased risk of death. We corroborate the clinical analysis by demonstrating experimentally that Δ133p53β promotes invasion using a panel of breast cancer cell models expressing WT and mutant *TP53* gene. Similar results were obtained in a panel of colon cancer cell lines. Depletion of endogenous mutant Δ133p53β prevents invasiveness despite unaltered strong expression of mutant TAp53α. Reciprocally, introduction of WT Δ133p53β promotes invasion of WT *TP53* cells devoid of endogenous Δ133p53β expression. Altogether, our data indicate that cell invasiveness is regulated by Δ133p53β irrespective of whether they harbor a mutant or a WT p53 gene.

Challenging the paradigm that WT *TP53* always acts as a tumor suppressor protein, our data show that the WT *TP53* gene can also promote invasiveness by encoding Δ133p53β. Hence, our data established that the cell decision is not exclusively determined by canonical full- length p53 protein (TAp53α), either mutant or WT, but by the modular protein complex of p53 isoforms, which reprogram epithelial cells to pro-metastatic cells.

## Results

### Association between Δ133p53 isoforms expression and breast cancer clinico-pathological subtypes

The association of Δ133p53 (α, β, γ) isoforms with prognosis has not previously been investigated in breast cancer. As there is no antibody specific of Δ133p53α, Δ133p53β or Δ133p53γ protein isoform, we developed a nested RT-PCR method that specifically detects and identifies Δ133p53α, Δ133p53β or Δ133p53γ mRNA variants in tumor samples. Briefly, the RT-PCR method is based on the previously described RT-PCR method (*Khoury et al., 2013*) but its sensitivity and specificity have been improved (A fully detailed protocol is provided in the 'Extended Experimental Procedure' section and the primer sequences are described in *Table 1*). It enables to specifically detect and identify expression of Δ133p53α, Δ133p53β or Δ133p53γ mRNA variants in tumor samples. This nested RT-PCR analysis allows thus to investigate the association of Δ133p53α, Δ133p53β or Δ133p53γ expression with the clinico-pathological markers and/or patient clinical outcome.

Total RNAs were extracted from 273 randomly selected primary breast tumors (*Figures 1A,B* and *Table 1*). Expression of Δ133p53α mRNA was detected in 97/273 (35.5%), Δ133p53β mRNA in 23/273 (8.4%) and Δ133p53γ mRNA in 56/221 (25.3%) of the breast tumors analyzed (*Table 2*). To verify that detection of Δ133p53β mRNA is associated with Δ133p53β protein expression, protein extracts from 8 large breast tumors were analyzed by SDS-PAGE and western-blot with KJC8, a beta p53 specific antibody (*Figure 1—figure supplement 2*). Samples identified as expressing Δ133p53β mRNA also express Δ133p53β at the protein level, confirming the RT-PCR analysis data and that Δ133p53β protein is expressed in tumor samples.

Δ133p53β or Δ133p53γ are tightly associated with Δ133p53α expression [21/23 (91.3%); 53/56 (94.6%), respectively]. Almost all tumors expressing Δ133p53β express Δ133p53α. However, only 25% of tumors expressing Δ133p53γ also express Δ133p53β (14/56, *Table 3*). The strong association of Δ133p53α with either the Δ133p53β or Δ133p53γ isoforms is consistent with the fact that the three mRNAs are initiated from the same promoter in intron-4 of the human *TP53* gene (*Bourdon et al., 2005*).

**Table 1.** Primers for specific amplification of Δ133p53 isoforms mRNAs by nested RT-PCR. The specific region (exon or intron) that each of the primers target is indicated (Ex: exon; Int: intron), the sequences corresponding to the exon junction are underlined. (F): Forward, (R): Reverse. PCR fragment sizes (bp) corresponding to p53 isoforms are also highlighted. Quality of reverse-transcription is assessed by quantitative RT-PCR amplification (SybrGreen) of actin and p53 mRNAs (primers are included)

| p53 mRNA variant | PCR | Primer name and targeted region | 5' – 3' sequence |
|---|---|---|---|
| All p53 mRNA | | For6 (F) (Ex6) | TTGCGTGTGGAGTATTTGGAT |
| | | Rev7 (R) (Ex7) | TGTAGTGGATGGTGGTACAGTCAGA |
| Δ133p53α | 1st | D133F1 (F) (Int4) | TAGACGCCAACTCTCTCTAG |
| | | Rev10 (R) (Ex10) | CTT CCC AGC CTG GGC ATC CTT G |
| | 2nd (670bp) | D133F2 (F) (Int4) | ACT CTG TCT CCT TCC TCT TCC TAC AG |
| | | RDNp53 (R) (Ex9/Ex10) | CTC ACG CCC ACG GAT CTG A |
| Δ133p53β | 1st | D133F1 (F) (Int4) | TAGACGCCAACTCTCTCTAG |
| | | Rev10 (R) (Ex10) | CTT CCC AGC CTG GGC ATC CTT G |
| | 2nd (700bp) | D133F2 (F) (Int4) | ACT CTG TCT CCT TCC TCT TCC TAC AG |
| | | p53β (R) (Ex9β) | TCA TAG AAC CAT TTT CAT GCT CTC TT |
| Δ133p53γ | 1st | D133F1 (F) (Int4) | TAGACGCCAACTCTCTCTAG |
| | | Rev10 (R) (Ex10) | CTT CCC AGC CTG GGC ATC CTT G |
| | 2nd (670bp) | D133F2 (F) (Int4) | ACT CTG TCT CCT TCC TCT TCC TAC AG |
| | | p53γ (R) (Ex9/Ex9γ) | TCGTAAGTCAAGTAGCATCTGAAGG |
| actin | | actin (F) | ATCTGGCACCACACCTTCTACAATGAGCTGCG |
| | | actin (R) | CGTCATACTCCTGCTTGCTGATCCACATCTGC |

We analyzed the association of Δ133p53α, Δ133p53β, and Δ133p53γ with the current clinico-pathological subtypes of breast cancer: luminal A (ER+, PR+, HER2-), luminal B (ER+, PR−, HER2-), luminal/HER2+ (ER+, PR+/-, HER2+), HER2 positive (ER-, PR-, HER2+), and triple negative (ER-, PR-, HER2-), according to the recommendations of St Gallen International expert consensus (*Goldhirsch et al., 2013*). None of the three isoforms were associated with histological cancer type, number of metastatic lymph nodes, tumor size or grade (*Table 4*) and neither Δ133p53α nor Δ133p53γ were associated with breast cancer subtypes. However, Δ133p53β was significantly less frequently expressed in HER2 positive and luminal/HER2+ tumors than in other breast cancer sub-types (2/69, Fisher's exact test, p≤0.042, *Figure 1C*). *TP53* mutations were identified in 64 tumors (64/260, 24.6%). Δ133p53β expression was significantly associated with *TP53* mutation status ($\chi^2$ = 5.625, p≤0.018, *Figure 1C*), whereas Δ133p53α and Δ133p53γ expression levels were not (*Table 2*). In all tumors, Δ133p53α, Δ133p53β or Δ133p53γ carried the same *TP53* gene mutation, indicating that all the corresponding mRNAs derive from cancer cells and not from normal stromal or inflammatory cells.

Altogether, these data indicate that almost all tumors expressing Δ133p53β and/or Δ133p53γ also express Δ133p53α. However, Δ133p53β is distinct from Δ133p53α and Δ133p53γ, in that it is less frequently detected in breast tumors overexpressing HER2 and more frequently expressed in tumors expressing mutant p53 than in tumors expressing WT p53. Therefore, Δ133p53α, Δ133p53β, and Δ133p53γ are not randomly expressed.

## Δ133p53β expression and prognosis in primary breast cancer

The association of Δ133p53α, Δ133p53β, and Δ133p53γ expression with cancer patient outcome (cancer progression/disease-free survival and overall survival/death) was investigated by univariate analysis of our cohort of breast tumors. Δ133p53β was associated with cancer progression (CP) and death (CP: $\chi^2$ = 5.953, p≤0.015, death: $\chi^2$ = 7.126, p≤0.008) (*Figure 1C*), while Δ133p53α and Δ133p53γ were not. This was further confirmed by Kaplan-Meier log-rank analyses (Kaplan-Meier log-rank test, CP: $\chi^2$ = 6.221, 1 df, p≤0.013; death: $\chi^2$ = 5.731, 1 df, p≤ 0.017, *Figures 1D,E,F*,

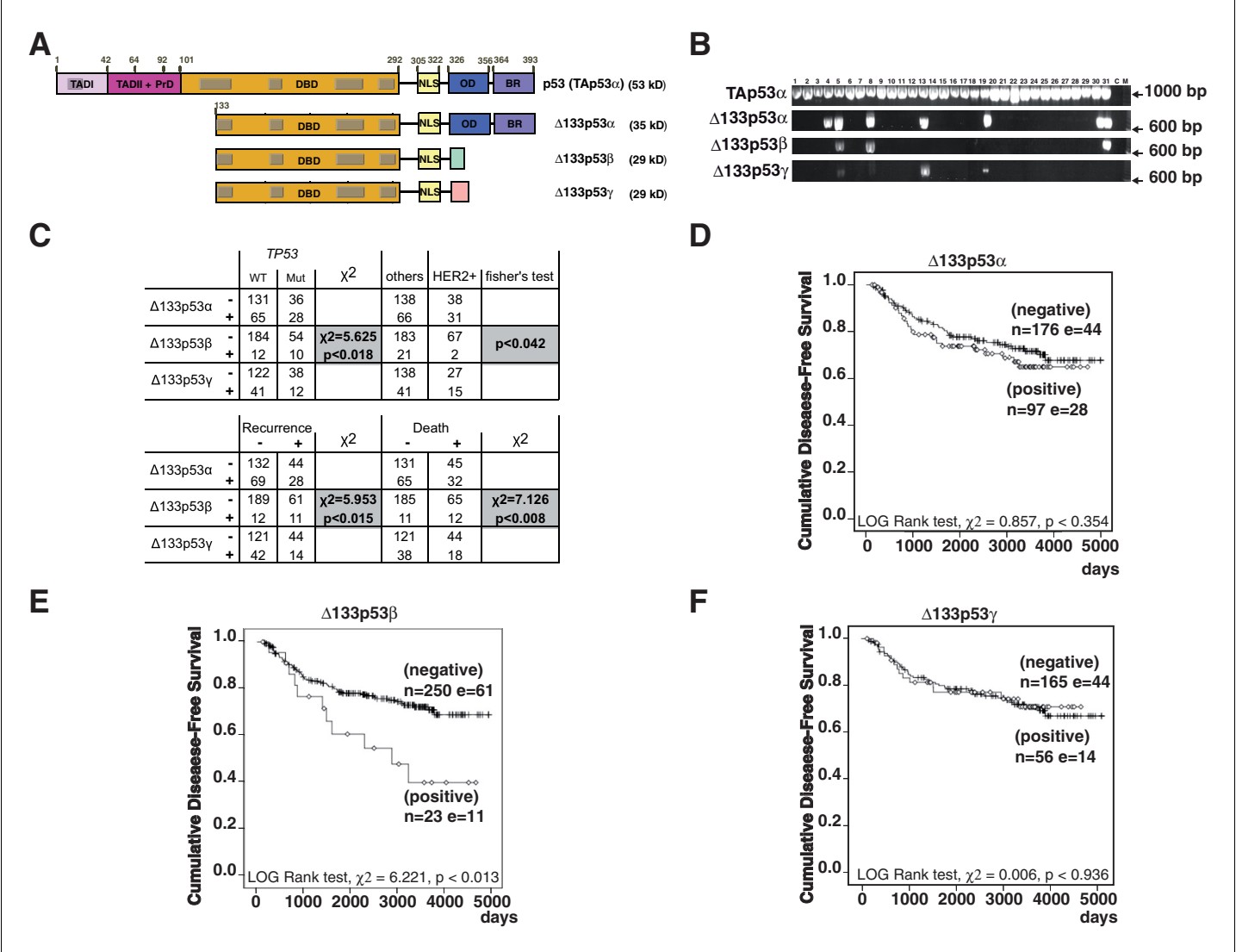

**Figure 1.** Breast cancer patients expressing Δ133p53β have a poor clinical outcome. (**A**) Schematic representation of human TAp53α, Δ133p53α, Δ133p53β and Δ133p53γ protein isoforms. The two transactivation domains [TADI (in light purple) and TADII (in pink)] the Proline-rich Domain (PrD), the DNA-Binding Domain (DBD in orange), and the C-terminal domain comprised of the nuclear localization signal (NLS; in yellow), the oligomerization domain (OD; in blue), and the basic region (BR; in violet) are represented. The grey boxes correspond to the five highly conserved regions defining the p53 protein family. The amino-acid positions defining the different p53 domains are indicated. The C-terminal domains of p53β (DQTSFQKENC) and p53γ (MLLDLRWCYFLINSS) are indicated by green and pink respectively. The molecular weight (kD) of each p53 isoform protein is indicated. (**B**) Specific nested RT-PCR amplification of Δ133p53α, Δ133p53β and Δ133p53γ. Total RNAs from 273 primary breast tumors were provided by the Tayside tissue bank. The quality of RNA and the quality of reverse-transcription were assessed as described in Materials and methods section. All samples with low RNA quality and low quality of reverse transcription were discarded to minimize the number of false negative. Each different p53 cDNAs were specifically amplified by 2 successive nested RT-PCR (35 cycles each) using 2 sets of the primer specific of each of the following p53 mRNA variants: p53 (all isoforms), Δ133p53α, Δ133p53β and Δ133p53γ. The nested RT-PCR analysis was performed as described in the Materials and methods section and the extended experimental procedures. Primer sequences are provided in *Table 1* . A representative subset of the nested RT-PCR analysis is shown. Tumor sample numbers are indicated. C: negative control, M: Molecular Markers. (**C**) Δ133p53β association with *TP53* gene mutation status, HER2 and clinical outcome in breast cancer patients. Δ133p53β expression is associated with p53 mutation but is not frequently expressed in HER2 positive tumors as determined by univariate analysis (upper table). Δ133p53β expression is associated with cancer progression and death in breast cancer by univariate analysis. (Lower table) (**D–F**) Only Δ133p53β is associated with cancer progression in breast cancer patients. Non-parametric Kaplan-Meier plots of disease-free survival in relation to Δ133p53α (**D**), Δ133p53β (**E**), Δ133p53γ (**F**) expression (n = 273). 'I' indicates censored cases on the curves. p-value is based on Kaplan-Meier log-rank analyses.

The following figure supplements are available for figure 1:

*Figure 1 continued on next page*

*Figure 1 continued*

**Figure supplement 1.** Non-parametric Kaplan-Meier plots analysis of marker association with disease-free survival and overall survival in the entire cohort of primary breast cancers (related to *Figure 1*).

**Figure supplement 2.** Detection of endogenous beta p53 protein isoforms in breast tumors(related to *Figure 1*).

*Figure 1—figures supplement 1A,B and C*). Importantly, $\Delta133p53\beta$ expression was also associated with cancer progression in WT *TP53* breast cancer patients (Kaplan-Meier log-rank test, CP: $\chi^2 =$ 5.232, p≤0.022) (*Figure 1—figure supplement 1I*), indicating that the association between $\Delta133p53\beta$ expression and cancer progression is not due to *TP53* mutation. (Of note the Kaplan-Meier log-rank analyses were also performed for $\Delta133p53\alpha$ or $\Delta133p53\gamma$ in WT *TP53* breast cancer patients. Their respective expression was not found associated with clinical outcome of WT *TP53* breast cancer patients.)

In addition to $\Delta133p53\beta$ expression, other well-established markers of cancer prognosis such as the breast cancer subtypes, *TP53* mutation, tumor grade, lymph node metastasis (absence or presence) and tumor size (>20 mm) were, as expected, also associated with cancer patient outcome in

**Table 2.** characteristics of breast tumors in the Tayside cohort.

**Variables**

| Primary breast tumors | | 273 |
|---|---|---|
| median age | | 61.5 (range 28.7–89.1 years) |
| follow up | | 6.85 (0.29–13.7 years) |
| grade | 1 | 25 |
| | 2 | 85 |
| | 3 | 159 |
| | unknown | 4 |
| type | ductal | 220 |
| | others | 53 |
| clinico-pathological subtype | triple-negatif | 49 |
| | luminal A | 112 |
| | luminal B | 43 |
| | HER2+ | 20 |
| | luminal/HER2+ | 49 |
| size | <20 mm | 91 |
| | >20 mm | 180 |
| | unknown | 2 |
| invaded lymph nodes | negatif | 132 |
| (node>0) | positif | 140 |
| | unknown | 1 |
| patient outcome | disease free | 201 |
| | recurrence | 72 |
| | alive | 196 |
| | death | 77 |
| p53 | Wild-type | 196 |
| | mutant | 64 |
| | unknown* | 13 |
| $\Delta133p53\alpha$ | - | 176 |
| | + | 97 |
| $\Delta133p53\beta$ | - | 250 |
| | + | 23 |
| $\Delta133p53\gamma$ | - | 165 |
| | + | 56 |
| | unknown* | 52 |

**Table 3.** Repartition of Δ133p53 isoforms in the Tayside breast cancer cohort (2x2 table). (A) Δ133p53α X Δ133p53β. (B) Δ133p53α X Δ133p53γ. (C) Δ133p53β X Δ133p53γ (−) not detected, (+) amplified.

**A**

|  |  | Δ133p53α | | |
|---|---|---|---|---|
|  |  | − | + | Total |
| Δ133p53β | − | 174 | 76 | 250 |
|  | + | 2 | 21 | 23 |
| Total |  | 176 | 97 | 273 |

**B**

|  |  | Δ133p53α | | |
|---|---|---|---|---|
|  |  | − | + | Total |
| Δ133p53γ | − | 138 | 27 | 165 |
|  | + | 3 | 53 | 56 |
| Total |  | 141 | 80 | 221 |

**C**

|  |  | Δ133p53γ | | |
|---|---|---|---|---|
|  |  | − | + | Total |
| Δ133p53β | − | 156 | 42 | 198 |
|  | + | 9 | 14 | 23 |
| Total |  | 165 | 56 | 221 |

our cohort (*Figure 1—figures supplement 1D–H*). To clarify the univariate analyses and adjust for possible confounding variables, the association of overall survival and disease-free survival with Δ133p53β, the breast cancer subtypes (luminal-A, luminal-B, HER2+, triple-negative and luminal/HER2+), *TP53* mutation, tumor grade, lymph node metastasis and/or tumor size (>20 mm) were investigated using the multivariate Cox's regression analysis. The cohort was composed of 112 luminal-A, 43 luminal-B, 49 luminal/HER2+, 20 HER2+ and 49 triple-negative tumors. We observed that the luminal-A subtype was, as expected, the most significant independent predictor of disease-free survival (HR = 3.09, 95% CI, 1.78 to 5.38, $p<1.10^{-4}$) and overall survival (HR = 4.15, 95% CI, 2 to

**Table 4.** Univariate analysis of Δ133p53 isoforms expression in relation to clinical pathological markers. Tumor grade, cancer type (ductal or others), tumor size (>or < 20 mm) were analyzed by Fisher's t-test. Association with the number of invaded lymph nodes were analyzed by Mann-Whitney method.

|  |  | grade | | | type | | | size | | | nb invaded Lymph nodes |
|---|---|---|---|---|---|---|---|---|---|---|---|
|  |  | 1-2 | 3 | *p* value | ductal | others | *p* value | <20 mm | >20 mm | *p* value |  |
| Δ133p53α | − | 76 | 98 | p<0.21 | 140 | 36 | p<0.56 | 62 | 113 | p<0.39 | p<0.97 |
|  | + | 34 | 61 |  | 80 | 17 |  | 29 | 67 |  |  |
| Δ133p53β | − | 102 | 144 | p<0.54 | 201 | 49 | p<0.8 | 81 | 167 | p<0.30 | p<0.60 |
|  | + | 8 | 15 |  | 19 | 4 |  | 10 | 13 |  |  |
| Δ133p53γ | − | 75 | 87 | p<0.25 | 133 | 32 | p<0.8 | 59 | 105 | p<0.79 | p<0.78 |
|  | + | 21 | 35 |  | 46 | 10 |  | 19 | 37 |  |  |

8.62, p<1.6 $10^{-4}$, *Table 5A*). Then, we examined by multivariate Cox's regression analysis the level of inter-dependence between *Δ133p53β*, *TP53* mutation, tumor grade, lymph node metastasis (absence or presence) and tumor size (>20 mm) in the luminal-A breast cancer population (n = 112). We determined by an Omnibus test of model coefficient that the Cox-regression model is fitted ($\chi^2$ = 17.589, 1 df, p<2.7 $10^{-5}$), indicating that the number of luminal-A tumors in our cohort is sufficient to support the conclusions. Among all the parameters included in the analysis (*Δ133p53β*, *TP53* mutation, tumor grade, lymph node metastasis and tumor size [>20 mm]), *Δ133p53β* stood out as the most significant independent predictor of cancer recurrence and death within the luminal-A sub-group regardless of *TP53* mutation status or presence of lymph node metastasis (Hazard ratio [HR], 7.93; 95% CI, 2.52 to 25; p<4.31 $10^{-4}$; [HR], 3.29; 95% CI, 1.125 to 9.63; p<0.03, *Table 5B*, respectively). It implies that at the time of surgery, detection of *Δ133p53β* in primary luminal-A breast tumors of patients devoid of lymph node would predict cancer progression. Of note, the multivariate Cox-regression analysis was also performed for *Δ133p53α* and *Δ133p53γ* but no significant association with the clinical outcome of luminal-A breast cancer patients was found.

Thus, *Δ133p53β* in luminal-A breast cancer, which predominantly expresses WT *TP53*, enhances on average by 8 times the risk of recurrence and by 3 times the risk of death in an otherwise excellent prognostic group. This indicates that WT *TP53* gene can be associated with cancer recurrence and death if it expresses *Δ133p53β*.

## p53 isoforms are more expressed in highly invasive breast cancer cells

The clinical study suggests that *Δ133p53β* may confer a more invasive phenotype to breast cancer cells. We then compared the ability of three different cancer cell lines to invade into type1- Collagen. We used: (1) luminal MCF7 cells (WT *TP53* and ER+), which depend on estrogen and EGF (Epidermal Growth Factor) for their growth and are neither locally invasive nor metastatic in mouse models, (2) the triple-negative MDA-MB-231 (mutant *TP53*-R280K, ER-), which was derived from a pleural effusion metastasis, and (3) its 'D3H2LN' variant (mutant *TP53*-R280K, ER-), which was selected for their enhanced tumor growth and widespread metastasis in mice (*Jenkins et al., 2005*).

As expected, MCF7 cells were very weakly invasive and MDA-MB-231 D3H2LN cells showed a significantly higher invasiveness as compared to parental MDA-MB-231 cells (*Figure 2A*). The

**Table 5.** Multivariate analysis of predictor. (A) Multivariate Cox's Regression analyses utilizing the forward step-wise elimination method to determine the degree of inter-dependence between the breast cancer subtypes (triple negative, Luminal A, Luminal/HER2 +, Luminal B, and HER2+), *Δ133p53β*, *TP53* mutation status, lymph node metastasis (present versus absent), tumor size and tumor grade in relation to disease-free survival and overall survival. (B) Multivariate Cox's Regression analyses utilizing the forward step-wise elimination method to determine the degree of inter-dependence between *Δ133p53βTP53* mutation status, lymph node metastasis (present versus absent), tumor size and tumor grade in the luminal A breast cancer patient population. The Fitted model was assessed by an Omnibus test. Hazard Ratio (HR), 95% confidence interval (CI), *p* values, number of iteration (itr) are indicated.

**A**

| n = 273 | | | omnibus test of model coefficients | | | | | | |
|---|---|---|---|---|---|---|---|---|---|
| | itr. | predictor | χ2 | df | p-value | HR | 95% CI | | p-value |
| Disease-free survival | 1 | Luminal A | 17.389 | 1 | 3.00E-05 | **3.09** | 1.78 | 5.38 | **1.00E-04** |
| Overall survival | 1 | Luminal A | 17.088 | 1 | 3.50E-05 | **4.15** | 2 | 8.62 | **1.60E-4** |

**B**

| n = 112 | | | omnibus test of model coefficients | | | | | | |
|---|---|---|---|---|---|---|---|---|---|
| | itr. | predictor | χ2 | df | p-value | HR | 95% CI | | p-value |
| Recurrence | 1 | Δ133p53β | 17.589 | 1 | 2.70E-05 | **7.93** | 2.52 | 24,96 | **4.31E-04** |
| Death | 1 | Δ133p53β | 5.31 | 1 | 2.10E-02 | **3.29** | 1.12 | 9.63 | **3.00E-02** |

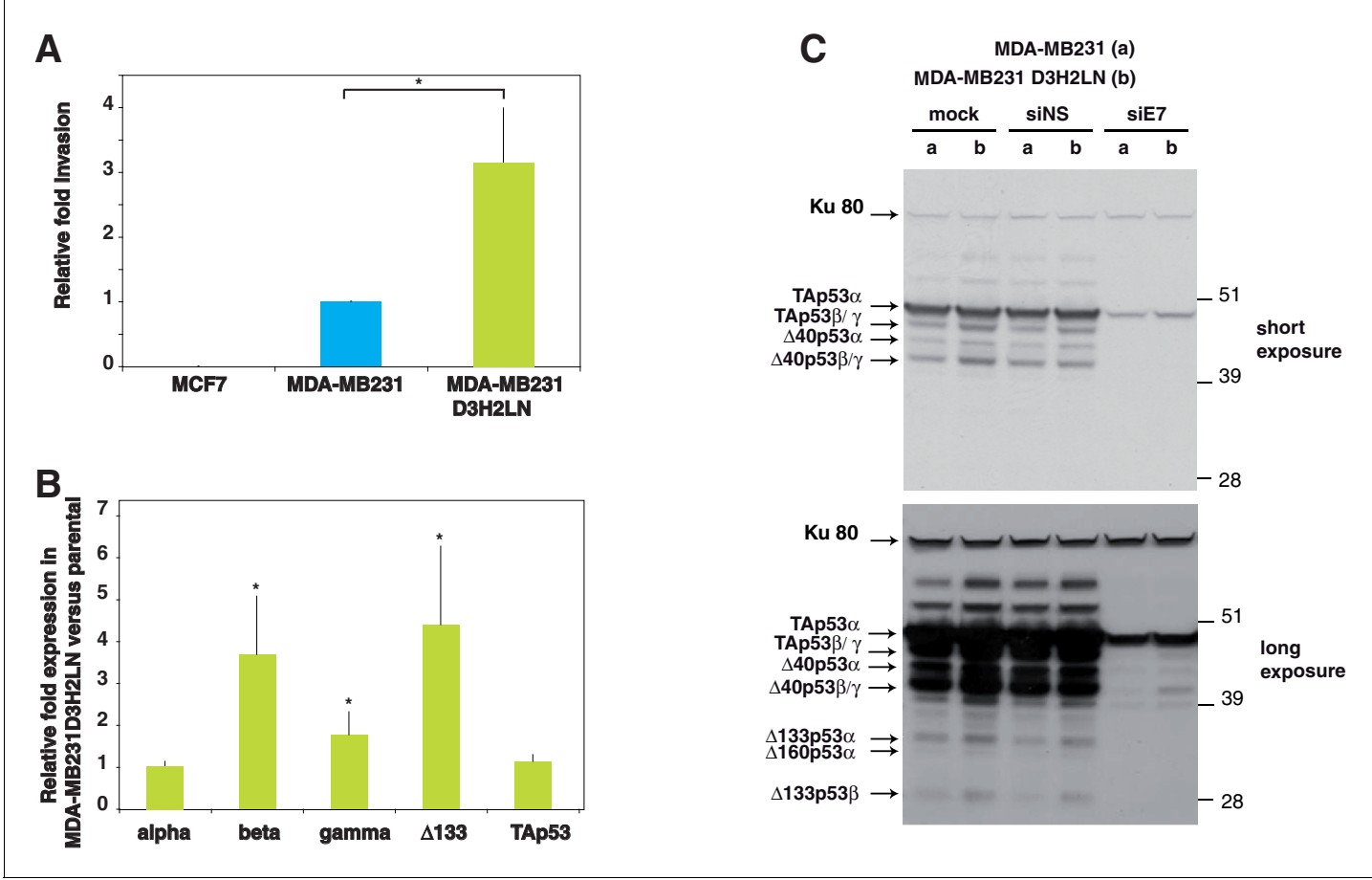

**Figure 2.** p53 isoform expression correlates with cell invasiveness in breast cancer cells. (**A**) Invasion of breast cancer cell lines. MCF7, MDA-MB231 and MDA-MB-231 D3H2LN cells were assayed for 24 hr and the changes in invasion were analyzed, as described in Materials and methods (*Smith et al., 2008*). Invading cells were counted as the number of invading cells at 50 μm divided by the number of non-invading cells at 0 μm. The results are expressed as the fold change ratio compared with MDA-MB231. Each assay was performed in triplicate for each cell line. The values plotted are means ± SEMs of N = 4 independent experiments; *p<0.05. (**B**) Comparative expression of p53 mRNA variants in MDA-MB231 D3H2LN versus parental MDA-MB231 cells. The expression level of p53 isoform mRNA is higher in the highly invasive MDA-MB231 D3H2LN cells than in MDA-MB231 cells. Subconfluent proliferating breast cancer cells were harvested for quantitative RT-qPCR Taqman assays (see Materials and methods section). p53 isoform expression was quantified relative to the control MDA-MB231 cell line. For all RT-qPCR experiments, expression levels of p53 isoforms were normalized to TBP. Results are expressed as the fold change compared to MDA-MB231 cells and represent means ± SEMs of N = 4 independent experiments; *p<0.05. (**C**) Differential expression of endogenous p53 protein isoforms in MDA-MB231 and MDA-MB231 D3H2LN cells. The protein expression levels of p53 isoforms were analyzed by western blotting using the sheep polyclonal p53 pantropic antibody (Sapu). To identify p53 protein isoforms, cells were transfected either with p53 siRNA (siE7) targeting exon-7 common to all p53 mRNA variants or with control siRNA (siNS, non specific). Two exposures (short and long) are shown.

difference between MCF7 cells and the two MDA-MB-231 cell lines may result from their *TP53* status, since mutant TAp53α protein can activate cell migration, invasion and metastasis (*Gadea et al., 2007, 2002*; *Muller et al., 2009*; *Roger et al., 2010*). However, the difference in invasion between the two MDA-MB-231 cell types expressing the same mutant *TP53* gene suggests the presence of another mechanism activated in the D3H2LN variant cell line. We examined whether this could be due to differential expression of the p53 isoforms. By quantitative real-time-PCR (RT-qPCR) using primers and probe (TaqMan) specific of the different subclasses of *TP53* mRNA variants TAp53, *Δ133* and alpha, beta and gamma (intron-9 splice variants) (*Khoury et al., 2013*), we found that whereas the two cell lines expressed similar levels of TAp53 and alpha *TP53* mRNA variants, the more invasive MDA-MB-231 D3H2LN expressed higher levels of *Δ133p53*, beta and gamma mRNA variants (*Figure 2B*). We confirmed, by western blotting, the highest expression level of p53 protein

isoforms in MDA-MB-231 D3H2LN cells by using a sheep polyclonal anti-p53 antibody (Sapu), which recognizes each p53 protein isoform with high specificity (*Marcel et al., 2013*). By western blotting, eight bands at 28, 35, 38, 40, 42, 45, 47, and 51 kDa were revealed in mock-transfected cells. (*Figure 2C*, mock lanes a and b). All these bands correspond to genuine p53 protein isoforms as they disappear after transfection with siRNA siE7 which targets exon-7 that is present in all p53 mRNA variants (lanes 'siE7') (*Aoubala et al., 2011*; *Camus et al., 2012*; *Terrier et al., 2012*). In agreement with the RT-qPCR data, TAp53α protein expression is similar between MDA-MB-231 and MDA-MB-231 D3H2LN, while all the other p53 protein isoforms, notably Δ133p53β are expressed at higher levels in MDA-MB-231 D3H2LN. This suggests that the enhanced invasiveness of MDA-MB-231 D3H2LN is not due to variation of mutant TAp53α protein expression level (*Figure 2C*, all lanes compare b to a).

## Δ133p53 isoforms control breast cancer cell invasion

To address the role of mutant Δ133p53 isoforms in the enhanced invasiveness of MDA-MB-231 D3H2LN cells, we analyzed cells depleted either of all Δ133p53 isoforms (by using the siRNAs si133-1 or si133-2, which specifically target the 5'UTR of all Δ133 mRNAs [*Aoubala et al., 2011*]), or of all β isoforms (by using siβ, [*Camus et al., 2012*; *Terrier et al., 2012*]), or of all p53 isoforms except Δ133p53 (α, β, γ) by using siTAp53, a siRNA targeting p53 mRNA splice variant containing *TP53* exon-2 (*Figure 3A*). We assessed knockdown efficacy by RT-qPCR and western blotting using the Sapu and KJC8 antibodies (*Figures 3D* and *Figure 3—figure supplement 1A*). Importantly, depletion of Δ133 or β p53 variants (by transfection of si133-1, si133-2 or siβ) significantly decreased the invasiveness of mutant *TP53* MDA-MB-231 D3H2LN cells without altering the expression of mutant TAp53α protein (*Figure 3D*). However, depletion of all p53 isoforms except Δ133p53 (α, β, γ) by siTA did not change the invasive activity of MDA-MB-231 D3H2LN cells, indicating that the simultaneous depletion of all TAp53 (including mutant TAp53α) did not impair MDA-MB-231 D3H2LN cell invasion (*Figures 3B* and *Figure 3—figures supplement 1A*). To confirm the role of the Δ133p53β isoform in promoting invasion potential, we re-introduced a si-resistant Δ133p53β-R280K isoform in MDA-MB-231 D3H2LN cells in which all Δ133p53 (α, β, γ) isoforms had been knocked down with si133-1 or si133-2. As expected, expression of the si133-1- and si133-2- resistant Δ133p53β-R280K rescued invasion (*Figure 3C* and *Figure 3—figure supplement 1B*). Similarly, Δ133p53β-R280K expression rescued invasion when cells were depleted of all β isoforms by siβ (*Figure 3C* and *Figure 3—figure supplement 1C*). We then compared the effects of Δ133p53α-R280K, Δ133p53β-R280K and Δ133p53γ-R280K isoforms on cell invasion. As the siRNAs si133-1 and si133-2 specifically target the 5'UTR of all Δ133 mRNAs, we tested their respective contribution to the invasive phenotype by re-introducing each of them in MDA-MB-231 D3H2LN cells in which all Δ133p53 (α, β, γ) isoforms had been knocked down with si133-1 or si133-2. All three mutant Δ133p53 isoforms rescued invasion in cells depleted of Δ133p53 variants, with Δ133p53β-R280K conferring the highest invasive activity compared to Δ133p53α-R280K or Δ133p53γ-R280K (*Figure 3C* and *Figure 3—figure supplement 1B*). These results indicate that mutant Δ133p53 isoforms, notably the Δ133p53β variant, are sufficient to promote invasion in mutant *TP53* MDA-MB-231 D3H2LN cells, consistently with an active role for Δ133p53β isoform in cancer progression.

EMT is a gatekeeper process involving global cell reprogramming leading to profound phenotypic changes that include enhanced invasiveness along with loss of epithelial cell-cell adhesion proteins such as E-Cadherin and concomitant gain of mesenchymal protein expression including Vimentin and N-Cadherin. EMT is a reversible process, thus mesenchymal cells can reverse their phenotype and re-express epithelial markers (*Thiery et al., 2009*). Interestingly, depletion of mutant Δ133 or β p53 isoforms reversed mesenchymal traits of MDA-MB-231 D3H2LN cells, by inducing a significant induction of E-Cadherin protein expression and a marked decrease in Vimentin expression, particularly in cells transfected with siβ. Knockdown efficacy was assessed by western blotting using the p53 pantropic polyclonal sapu and β-specific KJC8 antibodies (*Figure 3D*). Thus, inhibition of endogenous mutant Δ133 or β p53 isoforms induced a switch in the expression of these epithelial features which reflects global reversion of EMT cell reprogramming. Interestingly, re-epithelialization was impaired upon depletion of all p53 isoforms (*Figure 3D*, lane siE7), suggesting that other p53 splice variants control epithelial features. Differential induction of E-Cadherin expression was also observed at the mRNA level (*Figure 3E*).

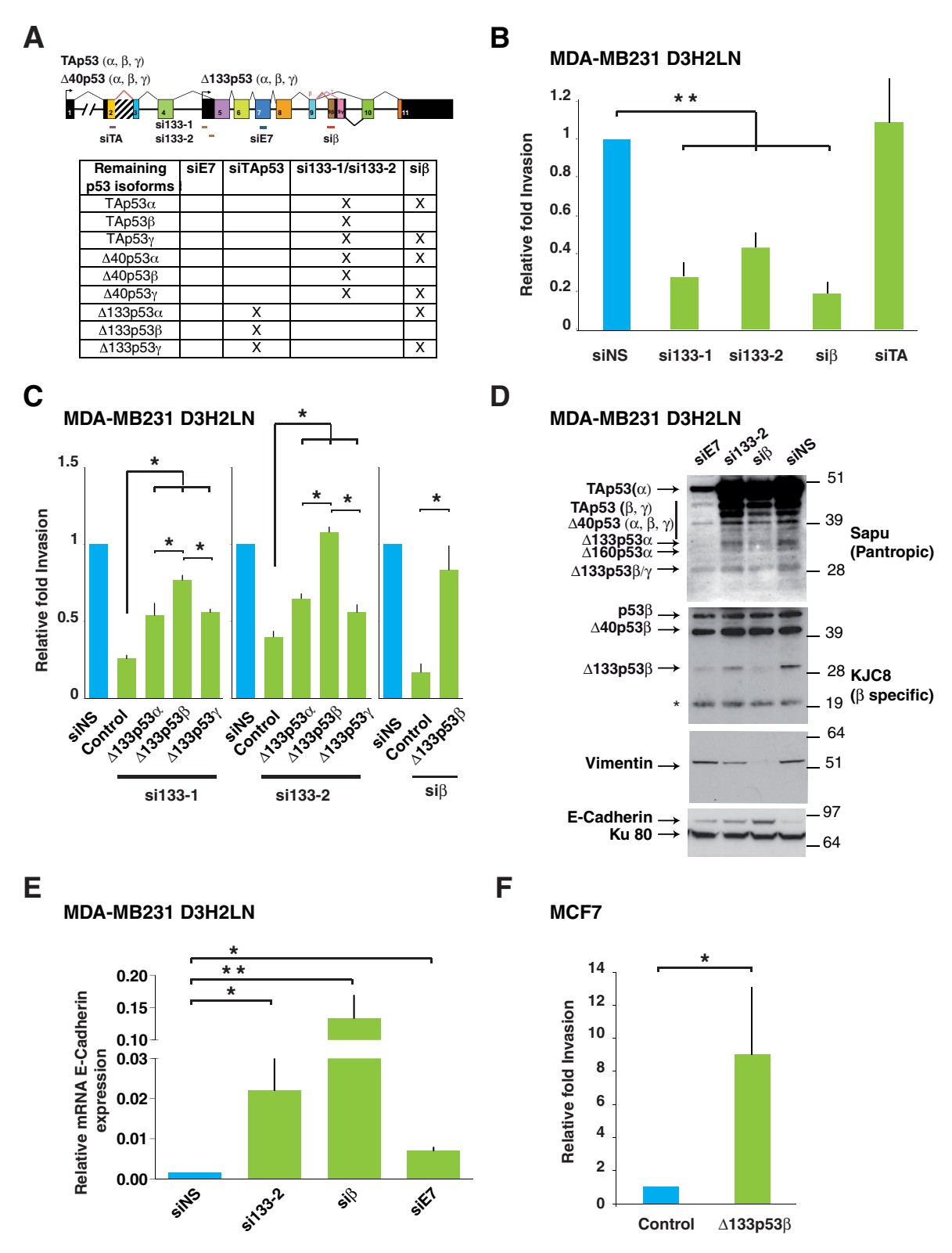

**Figure 3.** *Δ*133p53*β* isoform promotes invasion in breast cancer cells. (**A**) RNA chart of the different isoforms of *TP53* in this study showing the exons (boxes) and introns (horizontal lines, not to scale). Alternative promoters are shown as arrows and alternative splices are depicted with the lines above as they connect different exons. Location of the different siRNAs used is indicated below the chart. Below is a list of the p53 isoforms that remain after transfection of the different siRNAs, indicated by a cross. (**B**) Specific inhibition of some p53 isoforms expression decreases invasiveness. MDA-MB231

*Figure 3 continued on next page*

*Figure 3 continued*

D3H2LN cells were transfected with si133-1 or si133-2, two distinct siRNAs specific for the 5'UTR of Δ133p53 mRNAs; or with siTAp53, a siRNA targeting *TP53* exon-2 depleting all p53 isoforms except the Δ133p53 (α, β, γ) isoforms; or with siβ, a siRNA targeting the alternatively spliced exon-9β of *TP53*; or with siNS, a non-specific siRNA used as negative control. (**C**) ' Rescue' experiments. Re-introductions of si-133–resistant mutant Δ133p53α-R280K, Δ133p53β-R280K or Δ133p53γ-R280K restore the invasive activity in MDA-MB231 D3H2LN cells previously depleted of either Δ133p53 (α, β, γ) isoforms after transfection with si133-1 or si133-2, or β p53 isoforms (p53β, Δ40p53β and Δ133p53β) (n = 4). (**D**) Inhibition of endogenous p53 protein isoforms induces expression of epithelial features associated with decreased invasiveness. The expression of endogenous p53 protein isoforms after transfection of MDA-MB231 D3H2LN cells with si133-2, siβ, siE7 or control siRNA (siNS) was analyzed by western blotting using pantropic p53 isoforms antibody Sapu or KJC8, a β-specific p53 antibody recognizing p53β, Δ40p53β and Δ133p53β. The expression of two EMT markers (Vimentin and E-Cadherin) was determined in parallel. Ku80 was used as a loading control. * cross-reaction. (**E**) Quantification of E-Cadherin mRNA in MDA-MB231 D3H2LN cells transfected with siRNA si133-2, siβ, siE7 or siNS (control) used as a negative control. For all RT-qPCR experiments, expression levels were normalized to TBP. Results are expressed relative to TBP mRNA and represent means ± SEMs of N = 4 independent experiments; *p<0.05; **p<0.01 (**F**) WT Δ133p53β promotes cell invasion. Weakly invasive MCF7 cells were transfected with Δ133p53β expression vector or the empty expression vector (Control). Cells were challenged for their invasive potential after 48 hr. The values are plotted as means ± SEMs of at least 3 independent experiments; *p<0.05.

The following figure supplement is available for figure 3:

**Figure supplement 1.** Δ133p53 (α, β and γ) expression in MCF7 breast cancer cells (related to *Figure 3*).

We then addressed whether the introduction of WT Δ133p53β in MCF7 cells that endogenously express all p53 isoforms except Δ133p53β and Δ133p53γ (*Figure 3—figure supplement 1D*) promotes invasion. As shown in *Figure 3F* and *Figure 3—figure supplement 1E*, WT Δ133p53β triggered MCF7 cell invasion into type1-Collagen.

Altogether, the results in MDA-MB-231 D3H2LN and in MCF7 indicate that mutant and WT Δ133p53β promote invasion of WT and mutant *TP53* breast cancer cells. In mutant *TP53* cells expressing Δ133p53β, depletion of TAp53α does not inhibit cell invasion, while depletion of the Δ133p53 or β p53 isoforms decreases invasion of cells expressing mutant TAp53α. In WT *TP53* cells expressing all p53 isoforms except Δ133p53β, introduction of WT Δ133p53β promotes invasion. It suggests an active role for WT and mutant Δ133p53β in cancer progression corroborating the clinical analysis.

## Δ133p53 isoforms are required for colon cancer cell invasion

To determine whether Δ133p53β can promote invasion in other cancer types regardless of *TP53* mutation status, we studied a panel of WT and mutant *TP53* cell lines derived from colon carcinoma, another relevant model for cancer cell invasion, by comparing their ability to invade into Matrigel. The cell lines used were derived from the primary colorectal carcinoma (CRC): HCT116 (WT *TP53*) and SW480 (mutant p53R273H), and from metastatic tumors: LoVo (WT *TP53*), SW620 (mutant p53R273H) and CoLo205 (mutant p53, Y103 del27bp). Of note, the SW620 cell line was derived from a metastasis of the primary tumor from which the SW480 cell line was derived one year earlier. Compared to cells derived from primary colon tumors, metastasis-derived cell lines expressing WT or mutant *TP53* exhibited a higher ability to invade Matrigel (*Figure 4A*) and had no E-Cadherin at cell-cell junctions (*Figure 4B*). Of note, although the SW620 and SW480 cell lines have a mutated *TP53* gene (R273H) and come from the same patient, only SW620 are highly invasive, indicating that the expression of mutant TAp53α-R273H protein is not sufficient to explain invasive activities. We thus measured Δ133p53 mRNA expression by RT-qPCR (Taqman) and found it significantly higher in the invasive cell lines whether they express WT or mutant *TP53* gene, i.e. LoVo, SW620 and CoLo205, as compared with HCT116 and SW480 cell lines (*Figure 4C*), in agreement with the results obtained in the breast cancer cell lines. To confirm the role of Δ133p53 isoforms in cell invasion, we knocked-down these isoforms in mutant *TP53* SW620, or WT *TP53* LoVo CRC cells by using si133-1 and si133-2, which led to a significant decrease invasive capacity in both cell lines (*Figure 4D* and *Figure 4—figures supplement 1A and B*). As expected, re-introduction of si133-1-resistant WT Δ133p53β rescued invasiveness of LoVo cells depleted of Δ133p53 isoforms by si133-1 (*Figure 4—figures supplement 1C and D*). This indicates that Δ133p53 isoforms, notably Δ133p53β, regulate invasion, irrespectively of *TP53* mutation status.

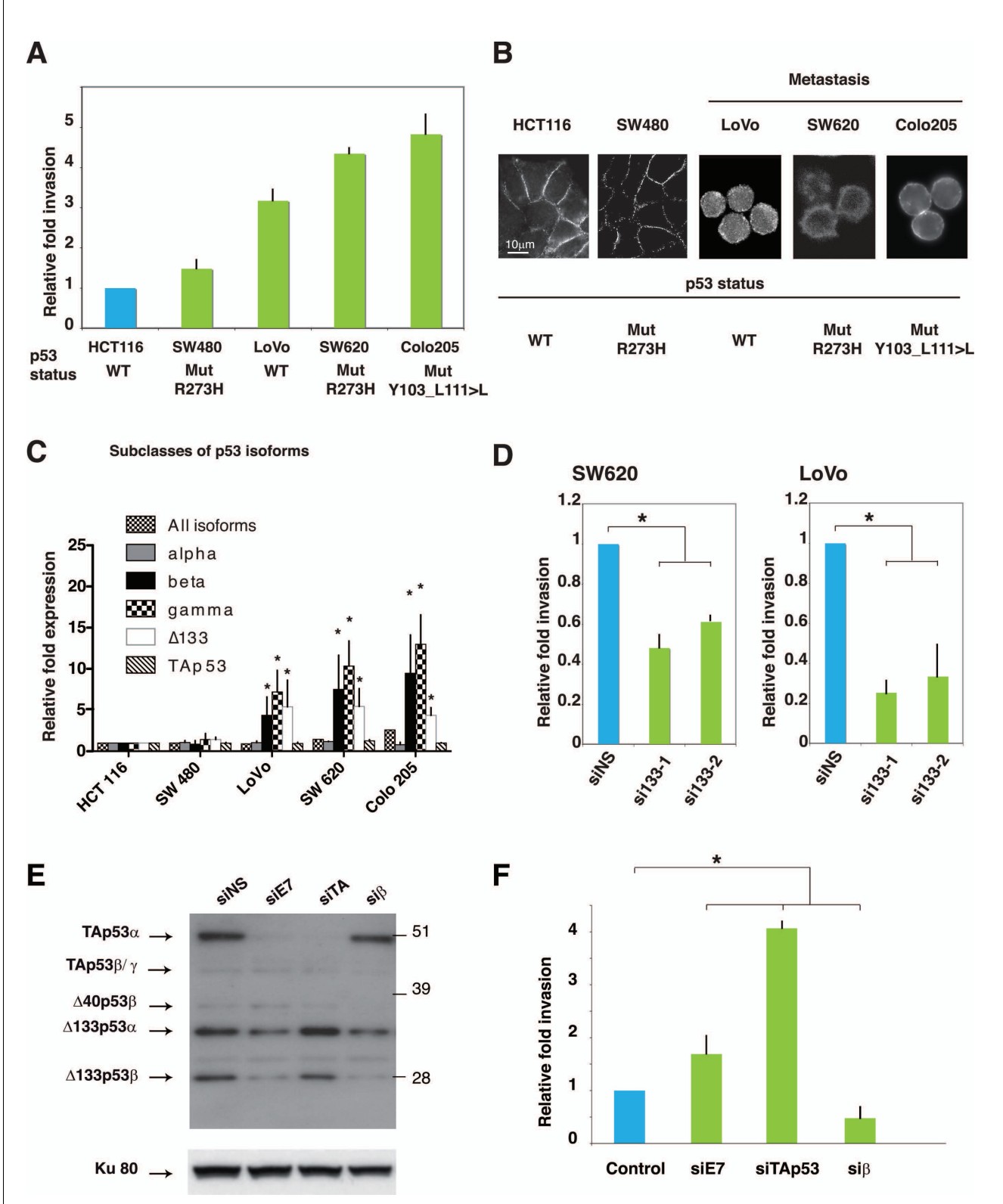

**Figure 4.** Δ133p53 isoforms promote invasion in colorectal cancer cells. (A) Invasiveness of different colorectal cancer cells. Cells were assayed for invasiveness through Matrigel over 24 hr, as described in Materials and methods. Invading cells were counted and the results are expressed as the average of 6 different fields and normalized to HCT116 cells. Each assay was performed in triplicate for each cell line. *TP53* mutation status is indicated. The values plotted are means ± SEMs of N = 4 independent experiments. (B) Immunofluorescence staining for E-Cadherin in a panel of colorectal cell

*Figure 4 continued on next page*

*Figure 4 continued*

lines. The images show representative E-Cadherin immunostaining for each colorectal cell line plated at low density. Scale bar: 10 µm. (C) Quantitative RT-qPCR (TaqMan) of different subclasses of p53 isoform mRNA in a panel of colorectal cell lines. Sub-confluent proliferating colorectal cells were harvested for quantitative RT-PCR assays (see Materials and methods section). For each cell line, results are expressed as the fold change compared to HCT116 cells. For all RT-qPCR experiments, expression levels of sub-types of p53 mRNA variants were normalized to TBP mRNA and represent means ± SEMs of N = 3 independent experiments; *p<0.05. (D) Invasion of SW620 (left) or LoVo (right) colon cancer cell lines after depletion of Δ133p53 (α, β, γ) isoforms. Cells transfected with Δ133 siRNAs (si133-1 and si133-2) or siNS were examined for their invasive potential after 24 hr. Invading cells were counted and the results are expressed as the average of 6 different fields and normalized to siNS. The values are plotted as means ± SEMs of 3 independent experiments; *p<0.05. (E) Western-blot analysis of endogenous p53 protein isoforms in HCT116 cells transfected with the non-specific siRNA (siNS) or with p53 isoform specific siRNA siE7, siTAp53, or siβ. Cells were then assessed for their invasive potential as in *Figure 4F*. (F) HCT116 cells transfected with siE7, siTAp53, siβ, or siNS were assessed for their invasive potential. Invading cells were counted and the results are expressed as the average of 6 different fields.

The following figure supplement is available for figure 4:

**Figure supplement 1.** Δ133p53 (α, β and γ) expression and effect on cell invasion using colorectal cancer cells.

Among WT *TP53* colon carcinoma cells, HCT116 are weakly invasive and endogenously express all p53 protein isoforms, including Δ133p53β (*Figure 4E*). We investigated whether we could modify HCT116 cell invasion by modulating the ratios of p53 isoforms. First, we depleted endogenous expression of different p53 isoforms using specific siRNAs and compared the invasiveness of HCT116 cells depleted of all p53 proteins other than Δ133p53 isoforms (i.e. transfected with the siTAp53 siRNA) or depleted of β isoforms but still expressing TAp53α (i.e. transfected with the siβ siRNA) (*Aoubala et al., 2011*; *Marcel et al., 2014*) (*Figure 4F*).

HCT116 cells transfected with siTAp53 are more invasive than cells transfected with siNS or siE7, while HCT116 cells transfected with siβ, which depleted p53β, Δ40p53β and Δ133p53β are less invasive than cells transfected with siNS (*Figure 4E and F*). This suggests that the ratio of TAp53 and β isoforms, including Δ133p53β, defines the invasive activity of HCT116 cells. Altogether our data indicate that the invasive capacity of colon cancer cell lines, as in the breast tumor cell lines, correlates with and depends on the expression of Δ133p53 isoforms, regardless of the *TP53* mutation status.

## Δ133p53β induces invasiveness and an amoeboid-like phenotype in WT *TP53* colon carcinoma cells

To determine whether the Δ133p53β-enhanced invasion directly impacts on the mode of cell migration, we transfected HCT116 with GFP- WT Δ133p53β and tracked them by video-microscopy. We observed two populations of cells, cohesive epithelial cells adherent to the glass and rounded cells loosely attached on top of cohesive cells. GFP-tagged Δ133p53β expression elicited a significant increase of rounded cells with dynamic bleb-like structures on their surface (*Figure 5A*, *Video 1* and *Figure 5—figure supplement 1A* as control). While recording, we noticed that some GFP-Δ133p53β -positive cohesive adherent cells detached progressively from the substratum, became rounded (white arrowhead, *Figure 5A*) and migrated (white arrow, *Figure 5A*). Propidium iodide staining and FACS analysis showed that blebbing cells were alive and were not undergoing apoptosis (*Figure 5—figure supplement 1B*). The number of blebbing cells was 25 times greater than the number of adherent cells in HCT116 cells transfected with Δ133p53β (*Figure 5B*). Similar rounded and detached cells were obtained using myc-tagged isoforms, ruling out any role of GFP in the observed phenotype (*Figure 5B*). Since loss of adhesive structures and concomitant acquisition of a rounded-blebbing movement are hallmarks of epithelial-amoeboid transition (EAT), a derivative of EMT which produce highly invasive cells, we further confirmed the EAT-like phenotype of Δ133p53β-expressing cells by measuring E-cadherin and β1-integrin levels, as the loss of both is a marker of EAT (*Friedl and Wolf, 2003*). As shown in *Figure 5C,E*-cadherin and β1-integrin, expressed at high levels in adherent cohesive cells, were barely detected in blebbing cells, implying they had experienced EAT. Δ133p53β also enhanced cell migration and invasion, since its expression in HCT116 cells elicited a two-fold increase in migration (*Figure 5D*) in uncoated transwell chambers and conferred a 5 times higher invasive capacity in Matrigel (*Figure 5D*) compared to GFP-only-transfected controls.

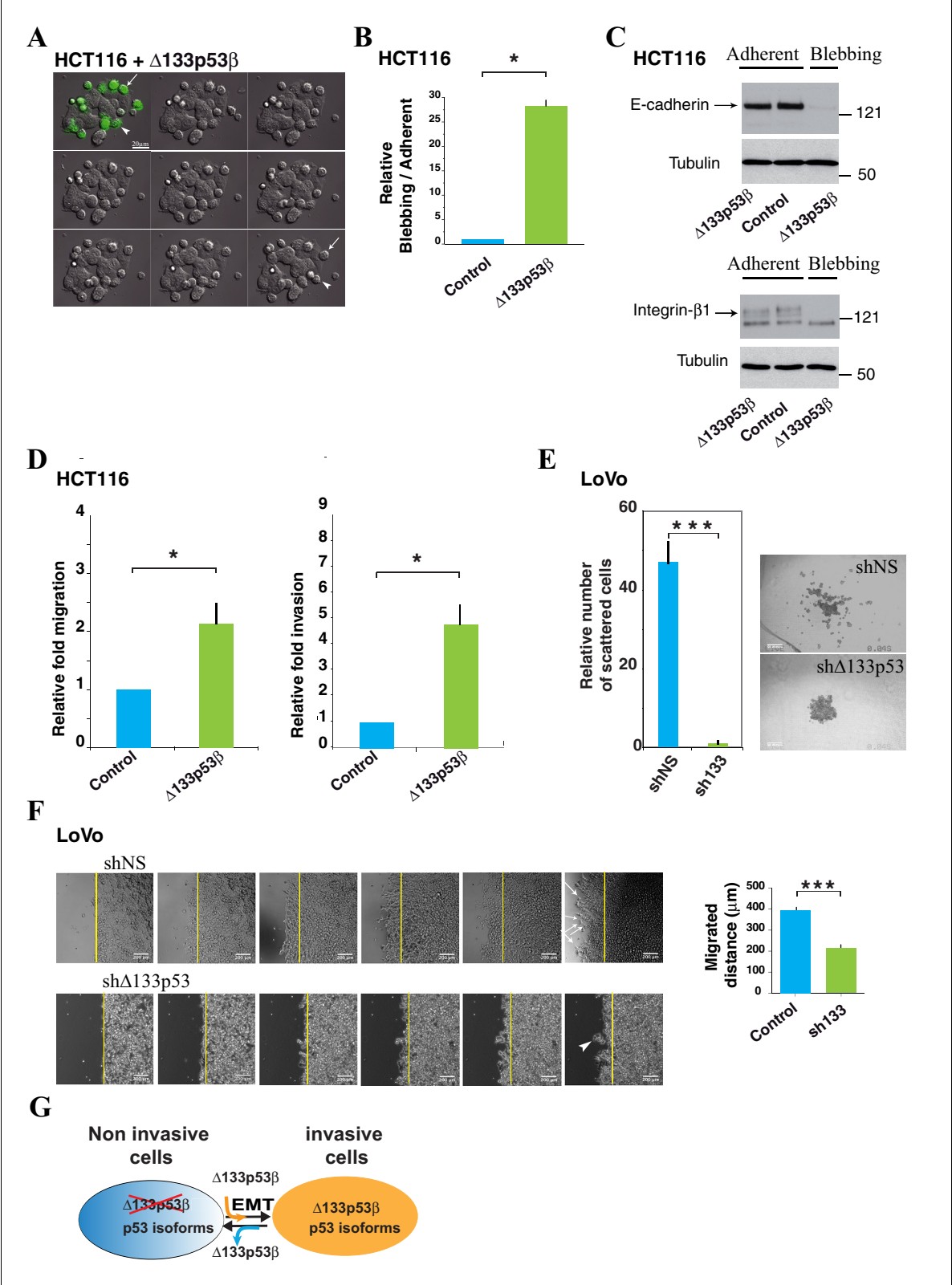

**Figure 5.** HCT116 cells expressing the *Δ133p53β* isoform display amoeboid-like movements. (**A**) Still time-lapse images of the accompanying video (Supplementary data, *video1*) of HCT116 cells transfected with the GFP-tagged *Δ133p53β* isoform. Cells were observed 48 hr after transfection. For the video, images were captured every 4 min during 12 hr. The panel represents 1/10 images i.e. one image every 40 min. *Δ133p53β*-transfected cells can be distinguished from non-transfected cells through expression of GFP (green). The *Δ133p53β*-transfected cells are rounded and exhibit blebbing
*Figure 5 continued on next page*

*Figure 5 continued*

movements on their surface. The arrow shows a cell that detaches from the others and from the dish during the time-lapse. The arrowhead shows a cell that still adheres to the other epithelial cells and to the substratum at the beginning of the experiment and then becomes progressively rounded. (**B**) Quantitative analysis of blebbing versus adherent cell number in the Myc positive cells. FACS analysis of the percentage of non-apoptotic blebbing Myc-positive HCT116 cells compared to total Myc-positive transfected with cells upon transfection of Myc-*Δ133p53β*, or Myc-empty expression vector (vector). Results were normalized to Myc-empty vector transfected cells. (**C**) Western blot analysis of the expression of E-Cadherin and *β*1-integrin in HCT116 cells expressing the GFP-tagged *Δ133p53β* isoform; Control: GFP-tag vector. Adherent: cells still adherent to the substratum; Blebbing: cells detached from the substratum and showing blebbing movements. Loading normalization was performed using an anti-α-tubulin antibody. (**D**) *Δ133p53β*-transfected HCT116 cells were quantified for their migration ability after 2 hr of migration through the Boyden chamber or for their invasiveness after 24 hr of the invasion through Matrigel, as indicated. The values are plotted as means ± SEMs of at least 3 independent experiments. (**E**) 3-D LoVo cell scattering. The numbers of scattered cells were quantified using Metamorph software (left). Cells were judged as « scattered » when individual cells or clusters of cells had lost contact with the main colony, as visualized (right). Values (means ± SEMs) were calculated from 4 independent experiments (n = 48) ***p<0.001. (**F**) Wound healing assay in LoVo cells infected with shRNA non relevant (shNS: shLuciferase) or sh*Δ133p53*. Cells were observed 25 hr after infection. Still time-lapse images of the accompanying videos (Supplementary data, *Videos 2* and *3*) For the videos, images were captured every 1 hr during 25 hr. The arrows show cells detaching from the others and migrating as individual cells. The arrowhead shows a cluster of cells which leaves the cohesive epithelium and which collectively migrate and enter into the gap. (**G**) Schematic representation of the role for *Δ133p53β* in reprogramming cells toward the invasive process. For WT or mutant *TP53* cells devoid of Δ133p53b expression, introduction of *Δ133p53β* promotes EMT and invasion. Reciprocally WT or mutant *TP53* cells expressing *Δ133p53β* have enhanced invasive activity. Depletion of *Δ133p53β* reverts EMT and inhibits invasion.

The following figure supplement is available for figure 5:

**Figure supplement 1.** Control experiments of *Δ133p53β* expression and effects in colon cancer cells.

Our data indicate that expression of WT *Δ133p53β* promotes EAT in WT *TP53* HCT116 cells which endogenously express low levels of *Δ133p53β*. We therefore next investigated whether we could modify EMT/EAT features in WT *TP53* CRC cells that endogenously express high levels of *Δ133p53β*. The WT *TP53* colon carcinoma LoVo cells are strongly invasive and endogenously highly express *Δ133p53* isoforms at the mRNA and protein levels (*Figure 4A,C*). Knockdown of *Δ133p53* isoforms in LoVo cells remarkably decreased 3-D cell scattering, a process which shares cellular features reminiscent of cells undergoing *EMT/EAT* (*Figure 5E* and *Figure 5— figure supplement 1C*). To investigate the impact of p53 isoforms-mediated cell scattering on the process of migration, we performed wound-healing assay. LoVo transduced with control shNS progressively detached and migrated as individual cells (arrows, *Figure 5F*). LoVo depleted of *Δ133p53* isoforms still remain cohesive and migrate very slowly into the gap. In this case, some rare clusters of cells leave the cohesive epithelium to enter into the gap (arrowheads, *Figure 5F*). Interestingly, the cluster of cells remains compact and cells move collectively, but with a reduced velocity compared to individual migrating cells with unaffected *Δ133p53* expression (*Figure 5F* and *Videos 2* and *3*). Thus endogenous *Δ133p53* protein isoforms regulate both the mode of cell motility and the ability to implement adherens junctions, with a subsequent impact on cell velocity.

Altogether, our data indicate that *Δ133p53β* expression elicits hallmarks of EAT/EMT in CRC cells.

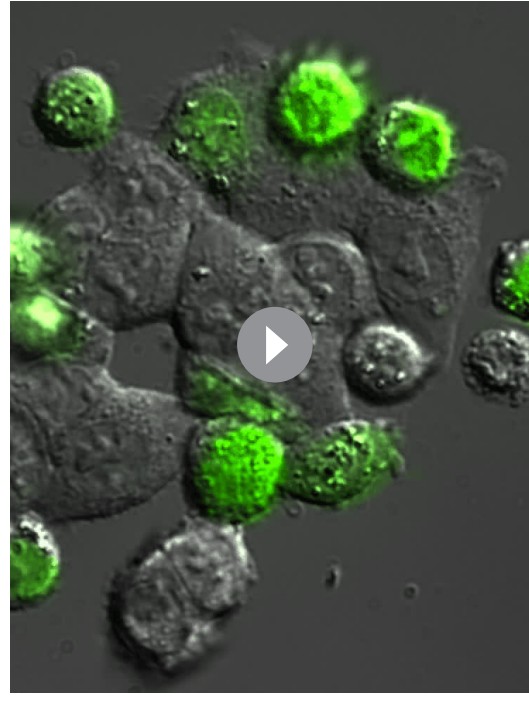

**Video 1.** (related to *Figure 5A*): DIC light microscopy of HCT116 cells expressing GFP-*Δ133p53β*.

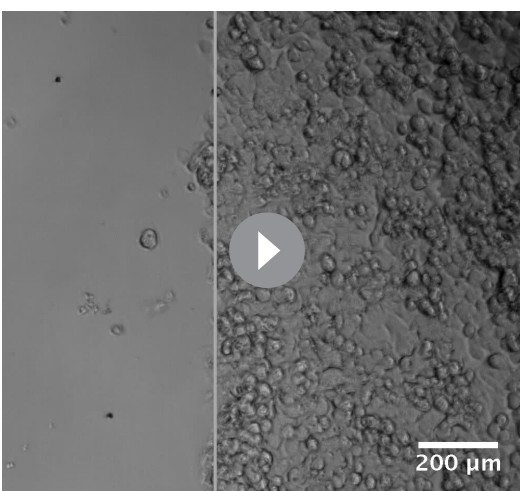

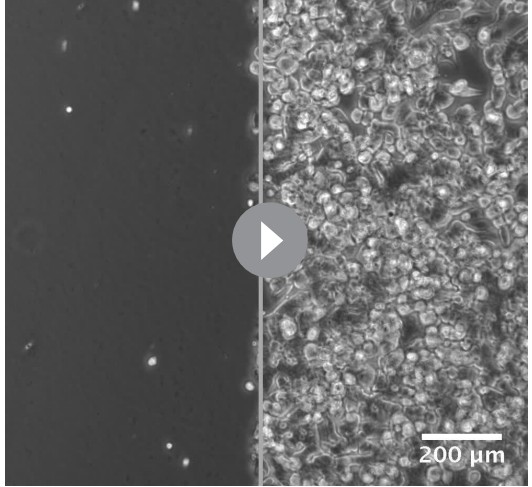

**Video 2.** (related to *Figure 5—figure supplement 1F*, upper panels).

**Video 3.** (Related to *Figure 5—figure supplement 1F*, lower panels): DIC light microscopy of Wound healing assay in LoVo cells infected with shRNA non relevant (shNS: shLuciferase; *Video 2*) or shΔ133p53 (*Video 3*). Cells were observed 25 hr after infection. Images were captured every 1 hr during 25 hr.

## Discussion

All the genetic evidence in the clinic and animal models indicate, unequivocally, the pivotal and fundamental role of the *TP53* gene in cancer formation, progression and response to treatment. *TP53* is ubiquitously expressed and it regulates different cell responses such as cell repair, proliferation, senescence, differentiation, cell migration and cell death in response to any change of tissue/cell homeostasis, thus maintaining tissue/cell integrity upon stress or damage. Consistently, *TP53* mutation is the most frequently mutated gene in human cancers and its mutation status is associated with poor clinical outcome. Any cancer treatments change cell homeostasis and therefore trigger *TP53*-mediated cell responses. It is currently impossible to accurately predict response to cancer treatment in the clinic because of the large variety of *TP53*-mediated cell responses. In addition, as missense mutations of *TP53* do not totally abrogate its tumor suppressor activity and can confer additional biological activities to *TP53* gene, it is very difficult to choose the most efficient cancer treatment based on *TP53* mutation status (*Kruiswijk et al., 2015*; *Lang et al., 2004*; *Li et al., 2012*; *Meek, 2015*). This indicates that major features of the *TP53* gene and its pathway have still to be uncovered and understood and this represents a critical bottleneck preventing major breakthroughs in cancer treatment.

To date, it is thought that the diverse biological activities associated with *TP53* gene expression are carried out by a single protein isoform, p53 (TAp53α). However, like most human genes, *TP53* gene does not express one but at least 12 different p53 protein isoforms which have distinct domain and activities.

The WT canonical p53 protein (TAp53α) prevents cancer invasion by inhibiting EMT (*Gadea et al., 2007*; *McDonald et al., 2011*; *Muller et al., 2011*; *Roger et al., 2010*; *Suva et al., 2013*) while the mutant canonical p53 protein (mutant TAp53α) was shown to promote cell invasion (*Adorno et al., 2009*; *Gadea et al., 2002*, *2004*; *Muller et al., 2009*; *Roger et al., 2010*; *Gadea et al., 2007*). However, despite this clear distinction between WT and mutant *TP53*, the situation in the clinic is not clear cut, as a mutant *TP53* expression in primary tumors does not systematically lead to metastasis formation and WT *TP53* is frequently expressed in metastasis (*Dong et al., 2007*; *Kalo et al., 2007*; *Oren and Rotter, 2010*).

Little is known about the biological roles and clinical relevance of p53 isoforms in cancer. In this study, we analyzed the expression of Δ133p53 isoforms (α, β, γ) in relation to clinical outcome in primary breast cancers. We determined that all tumors expressing Δ133p53β or Δ133p53γ also co-express Δ133p53α. This indicates that Δ133p53 (α, β, γ) isoforms are not randomly expressed in breast cancers, confirming that the internal promoter of *TP53* and the alternative (β, γ) splicing of

p53 mRNA are regulated in tumor cells (*Aoubala et al., 2011*; *Marcel et al., 2014*; *2010*; *Tang et al., 2013*).

We also determined that Δ133p53β expression is significantly associated with cancer recurrence in breast cancer patients, even in tumor expressing WT *TP53*. In particular, Δ133p53β enhances on average by eight times the risk of recurrence and by three times the risk of death in luminal A breast cancer patients, an otherwise excellent prognostic group. Importantly, detection of Δ133p53β at time of surgery in primary luminal-A breast tumors of patients devoid of lymph node would predict cancer progression. Therefore, far from having its function inactivated in Luminal A breast cancer, the WT *TP53* gene would promote breast cancer recurrence and increased risk of death when it expresses Δ133p53β.

To corroborate the clinical data, we experimentally investigated how Δ133p53β could confer cell invasion and motility to WT and mutant *TP53* breast cancer cells. First, we observed that the invasive activity of MCF7, MDA-MB-231 and MDA-MB-231 D3H2LN cells is positively correlated to the Δ133p53 mRNA expression level. This was also observed in a panel of WT or mutant *TP53* colon cancer cell lines. In mutant *TP53* MDA-MB-231 D3H2LN breast cancer cells, depletion of endogenous mutant Δ133p53 isoforms reduces cell invasion, despite unaltered and strong expression of mutant full-length p53 (TAp53α) protein. Re-introduction of si-133-resistant mutant Δ133p53α-R280K, Δ133p53β-R280K or Δ133p53γ-R280K in MDA-MB-231D3H2LN cells depleted of Δ133p53 isoforms restores their invasive phenotype, with mutant Δ133p53β-R280K being significantly the most potent. Importantly, depletion of the p53 protein isoforms with siTA that targets only the p53 proteins containing the transactivation domains (i.e TAp53α, TAp53β, TAp53γ, Δ40p53α, Δ40p53β, Δ40p53γ) without altering expression of Δ133p53 isoform did not change the invasive activity of MDA-MB-231 D3H2LN. This indicates that the simultaneous depletion of all TAp53 (including mutant TAp53α) did not impair MDA-MB-231 D3H2LN cell invasion.

It has previously been reported that overexpression of mutant TAp53α promotes integrin and epidermal growth factor receptor (EGFR) recycling, thus driving invasion (*Muller et al., 2009*). However, expression of the mutant TAp53α is not sufficient to explain why the triple negative breast cancer MDA-MB-231 and its highly metastatic MDA-MB-231 D3H2LN derived clone have different invasive activity while they both express the same mutant *TP53*-R280K gene. MDA-MB-231 D3H2LN cells showed a significantly higher invasiveness as compared to parental MDA-MB-231 cells. Interestingly, MDA-MB-231 D3H2LN expressed higher levels of Δ133p53 mRNA variants and proteins (*Figure 2B and C*). Similar results were observed in mutant *TP53* colon cancer cell lines. The SW480 and SW620 colon carcinoma cell lines derive respectively from the primary and secondary tumors resected from the same patient (*Hewitt et al., 2000*). Although SW480 and SW620 cells express the same mutant *TP53*-R273H gene, SW480 cells are far less invasive than the SW620 cells. Interestingly, SW480 cells express much less Δ133p53 mRNAs than the SW620 cells. Importantly, depletion of endogenous Δ133p53 isoforms from SW620 cells reduces cell invasion, despite unaltered and strong expression of mutant TAp53α protein. This indicates that the role of endogenous mutant Δ133p53 isoforms in promoting invasiveness may be extended to other tissue types and that endogenous Δ133p53 isoform expression regulates cancer cell invasion in *TP53* mutant colon and breast cancer cells, irrespective of the full-length p53 (TAp53α) expression.

Importantly, the invasive activity of Δ133p53 isoforms is not associated with *TP53* mutation status since depletion of Δ133p53 isoforms in WT *TP53* colon carcinoma LoVo cells abolishes their scattering and invasive activities. Δ133p53 isoform expression thus explains the inconsistent clinical association between *TP53* mutation and metastasis.

Interestingly, the introduction of WT Δ133p53β in the poorly invasive WT *TP53* MCF7 breast cancer cells that express all WT p53 isoforms but Δ133p53β, enhances MCF7 invasive activity. In WT *TP53* HCT116 colon cancer cells, that express all WT p53 protein isoform including low level of Δ133p53β protein, we demonstrated that HCT116 invasive activity can be controlled (enhanced or inhibited) by siRNAs specific of different p53 isoforms to modulate expression of different subsets of p53 protein isoforms. Hence HCT116 invasive activity is inhibited after depletion of Δ133p53 or β isoforms by siRNA si133 or siβ respectively while HCT116 invasive activity is enhanced after depletion of WT full-length p53 proteins (TAp53α, TAp53β, TAp53γ, Δ40p53α, Δ40p53β and Δ40p53γ) by siRNA siTA. Furthermore, ectopic expression of WT Δ133p53β in HCT116 cells enhances their invasive activity.

Altogether, Δ133p53β expression provides a rationale for invasive cancers expressing WT *TP53* and non-invasive cancers expressing missense mutant *TP53*. These data suggest that epigenetic mechanisms that stem Δ133p53β expression, i.e. alternative splicing and induction of the *TP53* internal promoter favour cancer invasion and dissemination of metastatic cells, regardless of *TP53* mutation status. Our data indicate that all three Δ133p53 isoforms (Δ133p53α, Δ133p53β and Δ133p53β) promote invasion. This is in accordance with recent reports showing that Δ133p53α stimulate angiogenesis and cell invasion (*Bernard et al., 2013*; *Roth et al., 2016*). Here we identified the Δ133p53β isoform as being the most efficient promoter of invasion, corroborating its role as a predictive indicator of cancer relapse and death in the clinic.

We investigated then how Δ133p53β regulate cell invasion in cancer cell lines. Since EMT leads to profound phenotypic changes due to genome-scale epigenetic reprogramming and post-transcriptional regulation, we studied the regulation of EMT by Δ133p53β to illustrate a molecular mechanism of Δ133p53β. Our data indicate that expression of Δ133p53β promotes acquisition of a rounded-blebbing movement, which is associated with decrease of E-cadherin and β1-integrin in colon cancer HCT116 cells. These phenotypic changes are hallmarks of epithelial-amoeboid transition (EAT), a derivative of EMT which produces highly invasive cells. Depletion of Δ133p53 isoforms also induces E-Cadherin expression and concomitantly inhibits Vimentin expression in invasive breast cancer MDA-MB-231 D3H2LN cells. Accordingly, knockdown of Δ133p53 isoforms in colon cancer LoVo cells decreased 3-D cell scattering, a process associated with *EMT/EAT.* Our study supports a critical role for Δ133p53β (whether WT or mutant) in the induction of cell invasion and EMT/EAT.

In summary, our data lead us to conclude that the biological activities associated with WT or mutant *TP53* gene expression are not carried out by the single full-length p53 protein (TAp53α). p53 isoforms underlie the dual role of WT *TP53* gene in preventing or promoting cancer cell invasion. The invasive activities of cancer cells expressing WT or mutant *TP53* gene can be enhanced or inhibited by respectively increasing or reducing expression of Δ133p53β (*Figure 5G*). Testing the prognostic discriminatory potential of Δ133p53β could influence clinical practice and may offer appropriate and yet unexplored therapeutic options for mutant or WT *TP53* tumors expressing Δ133p53β.

## Materials and methods

### Breast cancer patients

Primary, previously untreated and operable breast cancers from 273 Caucasian women, with sufficient tumor tissue surplus to diagnostic requirements and with complete clinical and pathological data, were analyzed. To minimize variations such as different surgical techniques and skills, which would have an effect on the rates of cancer recurrence and death, all tumors were provided by only one surgeon, Prof Alastair Thompson, breast cancer surgeon at Ninewells hospital in Dundee. Moreover, all patients were treated at Ninewells Hospital according to established clinical protocols. Furthermore, to minimize variation in tumor handling, all tumors were collected, stored, processed, extracted and stained by the Tayside Tissue Bank according to validated and standardized protocols at Ninewells Hospital. RNA quality was assessed using the BioAnalyzer 2100 prior to RT-PCR analysis and all samples with a ratio of 28S/18S < 1.2 were discarded. In addition, the Tayside Tissue Bank collected and verified the clinico-pathological data established by two anatomo-pathologists. Furthermore Tayside Tissue Bank collected and anonymised all the clinical data (treatment, patient follow-up).

The median age at diagnosis of the breast cancer patient cohort was 61.5 years (range 28.7 to 89.1 years) and median follow-up period was 6.85 years (range 0.29 to 13.7 years). Tumor tissues were macro-dissected by a specialist breast pathologist and snap frozen in liquid nitrogen prior to storage at −80°C. The samples were examined following Local Research Ethics Committee approval under delegated authority by the Tayside Tissue Bank (www.taysidetissuebank.org).

### Tumor grade, estrogen receptor, progesterone receptor and HER2 status

Immunohistochemical staining was carried out on 4 µm sections of formalin-fixed paraffin-embedded tumors, as previously described (*Purdie et al., 2010*).

## RT-PCR and p53 mutation analysis

To generate cDNA from tumor total RNA extracts, two independent reverse transcription of 300 ng of total RNA were performed and pooled (*Khoury et al., 2013*). The quality of the reverse Transcription (cDNA) was assessed by quantification of the Actin and p53 cDNAs using quantitative Real-Time PCR (Sybr Green) previously described (*Khoury et al., 2013*). All cDNA samples for which Actin or p53 cDNA could not be detected in less than 30 cycles were discarded as it indicates that the reverse-transcription did not work properly. This step reduces the number of false-negative samples that would bias the statistical analysis.

The sensitivity and specificity of the RT-PCR analysis were improved by performing 2 successive PCR of 35 cycles each and by using 2 sets of nested primers for each Δ133p53 mRNA variants. Therefore the RT-PCR analysis specifically amplifies Δ133p53α, Δ133p53β or Δ133p53γ mRNA. The primer sequences are provided in *Table 1 and a* detailed protocol is described in the 'extended experimental procedures'.

In the first PCR of 35 cycles, all Δ133p53 mRNA variants were amplified using a forward primer corresponding to the 5'UTR of the Δ133p53 mRNA (TP53 intron-4) and a reverse primer corresponding to the exon-10. Then, the product of the first PCR (1ul) is re-amplified in the second PCR of 35 cycles using a nested forward primer located in the 5'UTR of the Δ133p53 mRNA (intron4) and a nested reverse primer spanning the exon junction between exon-9 and exon-10 to specifically amplify the DΔ133p53α cDNA. To specifically amplify the Δ133p53β cDNA, the product of the first PCR (1ul) is re-amplified in the second PCR of 35 cycles using the nested forward primer located in the 5'UTR of the Δ133p53 mRNA (intron4) and a nested reverse primer corresponding to the exon9b encoding the β C-terminal amino-acid sequence. To specifically amplify the Δ133p53γ cDNA, the product of the first PCR (1ul) is re-amplified in the second PCR of 35 cycles using the nested forward primer located in the 5'UTR of the Δ133p53 mRNA (intron4) and a nested reverse primer spanning the exon junction between exon-9 and exon-9g that encodes the γ-C-terminal amino-acid sequence. The RT-PCR analysis was tested in p53-null cells (H1299) and on a panel of cell lines expressing *TP53* gene (MCF7, MDA-MB-231, HCT116). The products of the second PCR were then validated by sequencing and after electrophoresis on 1% agarose gel.

For the RT-PCR analysis of the tumor samples, the products of the second PCRs were validated for all tumors by electrophoresis on 1% agarose gel. The PCR products of at least 2 tumor samples were sequenced to validate identity of the amplified Δ133p53 mRNA variants. The samples giving rise to a PCR product of the expected size were deemed positive for the expression of the corresponding Δ133p53 mRNA variant otherwise they were deemed negative. The two successive PCR of 35 cycles using two nested sets of primer maximize sensitivity and specificity of amplification. It enables to detect and identify the expression of each Δ133p53 variants even if expressed in a small population of tumor cells, taking account thus of tumor cell heterogeneity. It allows thus to compare their expression and respective association with the clinico-pathological markers and/or patient clinical outcome. The expression of p53 isoforms and analysis of p53 mutation were performed as described in the extended experimental procedures.

## Statistical analysis

The statistical clinical analysis was performed as described previously (*Bourdon et al., 2011*). All statistical comparisons were made using Wilcoxon test and a *p*-value < 0.05 was considered to be statistically significant.

## DNA constructs, reagents and antibodies

Human p53 isoform constructs (*Bourdon et al., 2005*) were sub-cloned into pEGFPC1 (Clontech) or pLPCmyc. Antibodies were purchased from BD-transduction laboratories, Santa Cruz and GE-Healthcare. Dilutions are indicated in extended experimental procedures.

## Cell culture, transfections and adhesion assay

Cells were cultured at 37°C in the presence of 5% $CO_2$ in McCOY'5A or DMEM media (Sigma) for colon cancer and breast cancer cells, respectively. HCT116 (RRID:CVCL-0291), MDA-MB231 (RRID: CVCL-0062), MCF7 (RRID:CVCL-0031), LoVo (RRID:CVCL-0399), SW480 (RRID:CVCL-0546), SW620 (RRID:CVCL-0547), Colo205 (RRID:CVCL-0218) were purchased from ATCC. MDA-MB231 D3H2LN

(RRID:CVCL-D257) were purchased from PerkinElmer. All cell lines were tested for mycoplasma contamination after thawing using Lonza mycoalert assay (Reference number: LT27-236). Transfections of p53 isoforms and siRNA were carried out using JetPEI kit (Qbiogen) and Interferin (Polyplus), respectively, according to the manufacturers' instructions. For adhesion assays, non-adherent and adherent cells were collected 24 hr after transfection and counted using the Countess cell counting system (Invitrogen).

### Time-lapse imaging
Time-lapse DIC microscopy was performed on a Leica DMIRE2 inverted microscope with an automatic shutter and GFP filter sets, as described in the extended experimental procedures.

### Cell extracts, western blotting
Non-adherent and adherent cell extracts were obtained and analyzed by western blotting. Detailed protocols are described in the extended experimental procedures. Cells were lysed and protein complexes were co-immunoprecipitated using GFP-Trap beads (Chromotek) as previously described (*Arsic et al., 2012*). Sources of antibodies are provided in the extended experimental procedures.

### Migration and invasion assays
Colon cancer cell migration and invasion were performed using Boyden chambers (*Vinot et al., 2008*). Breast cancer cell invasion assays were performed as previously described (*Smith et al., 2008*).

### Quantitative Real-Time PCR (RT-qPCR or Taqman)
Total RNA was extracted with the RNeasy Mini Kit (Qiagen) and treated with DNase (Qiagen) prior to reverse transcription, which was carried out using oligo(dT) (Invitrogen) and M-MLV reverse transcriptase (Invitrogen). Sub-classes of p53 mRNA isoforms were quantified by Real-Time PCR (TaqMan) as previously described (*Aoubala et al., 2011*; *Moore et al., 2010*). All measurements were normalized to the expression of the TATA box-binding protein (*TBP*) gene.

### Extended experimental procedures
#### Tumor grade, estrogen receptor, progesterone receptor and HER2 status
Immunohistochemical staining was carried out on 4 μm sections of formalin-fixed paraffin-embedded tumors with the mouse monoclonal anti-estrogen receptor alpha (ER) antibody 6F11 (Novocastra Laboratories Ltd), progesterone receptor (PR) antibody clone 16, (Novocastra Laboratories Ltd) and mouse monoclonal anti-HER2 antibody CB11 (Novocastra Laboratories Ltd). Additional analyses were performed according to histological tumor grade (*Bloom and Richardson, 1957*) and graded by a specialist consultant breast pathologist. ER status (as ER negative 0–3 versus ER positive 4–18) was determined by the Quickscore method (*Detre et al., 1995*). HER2 scoring was performed as previously described (Purdie et al.,2010).

#### RT-PCR analysis
Approximately 10 mg of tumor tissue (>40% of tumor cells) was homogenized in 750 μl QIAzol lysis reagent (Qiagen Ltd, Crawley, West Sussex, UK) and total RNA was extracted (Qiagen). RNA quality was assessed using the BioAnalyzer 2100 (Agilent Technologies, Palo Alto, CA, USA), prior to RT-PCR analysis and all samples with a ratio of 28S/18S < 1.2 were discarded. To generate cDNA from tumor total RNA extracts, two independent reverse transcription of 300ng of total RNA were performed using the Cloned AMV Reverse Transcription kit (Invitrogen) following manufacturer recommendation (*Khoury et al., 2013*). The quality of the reverse Transcription (cDNA) was assessed by quantification of the Actin and p53 cDNAs using quantitative Real-Time PCR (Sybr Green) previously described. Primers are provided in *Table 1* (*Khoury et al., 2013*). All cDNA samples for which Actin or p53 cDNA could not be detected in less than 30 cycles were discarded. In the first PCR of 35 cycles, all Δ133p53 mRNA variants were amplified using a forward primer corresponding to the 5'UTR of the Δ133p53 mRNA (TP53 intron-4) and a reverse primer corresponding to the exon-10. Per reaction mix (1 sample): 5x Green buffer Go Tag (Promega): 10 μl; dNTP (10 mM): 1 μl; forward Primer (10 uM): 1 ul; reverse Primer (10 μM) 1 μl; Go Taq DNA (Promega) 0.25 μl; 2 μl of cDNA

complete at 50 µl final volume with milliQ water.Thermal cycling conditions for PCR amplification (1st PCR reactions) for p53 isoforms are as follows: [94°C for 3 min (1 cycle)], [94°C for 30 s, 63°C for 30 s, 72°C for 1 min 30 s (35 cycles)] and finally [72°C for 8 min (1 cycle)].

Then, the product of the first PCR (1ul) is re-amplified in the second PCR of 35 cycles using a nested forward primer located in the 5'UTR of the Δ133p53 mRNA (intron4) and a nested reverse primer spanning the exon junction between exon-9 and exon-10 to specifically amplify the D133p53α cDNA. To specifically amplify the Δ133p53β cDNA, the product of the first PCR (1 ul) is re-amplified in the second PCR of 35 cycles using the nested forward primer located in the 5'UTR of the Δ133p53 mRNA (intron4) and a nested reverse primer corresponding to the exon-9β encoding the β C-terminal amino-acid sequence. To specifically amplify the Δ133p53γ cDNA, the product of the first PCR (1ul) is re-amplified in the second PCR of 35 cycles using the nested forward primer located in the 5'UTR of the Δ133p53 mRNA (intron4) and a nested reverse primer spanning the exon junction between exon-9 and exon-9γ that encodes the γ-C-terminal amino-acid sequence. Per reaction mix (1 sample) : 10 µl; dNTP (10 mM): 1 µl; forward Primer (10 uM): 1ul; reverse Primer (10 µM) 1 µl; Go Taq DNA (Promega) 0.25 µl; Add 1 µl of c-DNA from1st PCR; complete at 50 µl final volume with milliQ water. Thermal cycling conditions for PCR amplification (2nd PCR reactions): [94°C for 3 min (1 cycle)], [94°C for 30 s, 60°C for 45 s, 72°C for 1 min 30 s (35 cycles)] and finally [72°C for 8 min (1 cycle)].

The PCR products are then analyzed by electrophoresis on 1% agarose gel. The RT-PCR analysis was tested in p53-null cells (H1299) and on a panel of cell lines expressing *TP53* gene (MCF7, MDA-MB-231, HCT116). The products of the second PCRs were validated for all tumors by electrophoresis on 1% agarose gel. The PCR products of at least 2 tumor samples were sequenced to validate identity of the amplified Δ133p53 mRNA variants.

### TP53 mutation analysis
*TP53* mutation status was determined from p53 PCR fragment as previously described (*Bourdon et al., 2005*).

### Statistical analysis
The primary outcomes in this study were breast cancer-specific overall survival (abbreviated to overall survival) and breast cancer-specific disease-free survival (abbreviated to disease-free survival or cancer recurrence throughout the text), and accordingly, non-breast cancer deaths were censored at the time of death (i.e. at the time of their death, the women were considered to have survived breast cancer but died of other causes).

### DNA constructs, reagents and antibodies
The human p53 isoform constructs used in this report were previously described (*Bourdon et al., 2005*). They were sub-cloned into the EcoRI and BamHI sites of pEGFPC1 (Clonetech) to obtain GFP-tagged proteins or in the BamHI and EcoRI sites of pLPCmyc to obtain myc-tagged proteins. Constructs were amplified using the Nucleobond PC 500 kit (Macherey-Nagel) according to the manufacturer's instructions. Human Δ133p53 isoforms (α, β and γ) were cloned into pMSCVhyg (Clontech Laboratories) plasmid for retrovirus production. pSIREN-Luc (luciferase)-shRNA was purchased from Clontech. The mouse anti-E-Cadherin (clone 36) and the mouse anti-Beta1-Integrin, were purchased from BD-transduction laboratories and were diluted at 1/400° and 1/250°, respectively. The mouse N-Cadherin (clone 32/BD Transduction laboratories), and the rabbit polyclonal anti-GFP (Invitrogen, Life Technologies, A-6455) were used at 1/400°, 1/250°, 1/300° and 1/2000°, respectively.

The sheep pantropic p53 antibody Sapu was diluted at 1/5000°. The rabbit polyclonal KJC8 antibody specific of the β p53 isoforms was diluted at 1/4000° (*Bourdon et al., 2005*). The Horse-Radish Peroxidase (HRP)-conjugated anti-IgG antibodies were purchased from GE-Healthcare and were diluted at 1/5000°. The Western Lightning Chemiluminescence (ECL) reagents were purchased from PerkinElmer.

## Time-lapse imaging

Time-lapse DIC microscopy was performed with a 63x oil-immersion objective (HC x PL APO 1.32–0.6 oil CS), sample heater (37°C) and homemade $CO_2$ incubation chamber. Images were captured with micromax CCD camera (1300Y/HS) imaging software, converted into TIFF files and edited and compiled with Metamorph software. The exposure time was 500 ms for GFP and 300 ms for light. Images were captured every 3 sand during 5 min or every 4 min during 12 hr.

## Cell extracts, western blotting

Media containing blebbing cells was spun down at 1200 rpm for 5 min and the pellet constituted of blebbing invasive cells was lysed. The remaining adherent cells were gently scraped in lysis buffer. The two populations of cells were analyzed separately for each condition (see Figure legends). Total protein concentration was determined using the BCA kit (Promega). Protein samples were electrophoretically separated on 8% SDS-PAGE gels for E-Cadherin and Beta1-Integrin. An equal amount of total protein (30 µg) was loaded into each lane. Proteins were then transferred onto nitrocellulose membranes. Membranes were blocked with TBS/0,1% Tween 20 containing 3% milk for one hour and then incubated overnight with the primary antibodies diluted in TBS/0,1% Tween 20 containing 3% milk. After several washes in TBS/Tween, membranes were incubated with anti-rabbit or anti-mouse Ig antibodies linked to HRP. Membranes were developed with ECL according to the manufacturer's instructions. All scanned autoradiographs were quantified using AIDA/2D densitometry software.

## Western blotting analysis of tissue samples

Tissues were dissected and snap frozen in liquid nitrogen. Tissue chunks were placed in an Eppendorf tube containing RIPA buffer (1% NP-40, 0.5% sodium deoxycholate, 0.1% SDS and PBS, pH: 7.4, supplemented with protease inhibitor Complete (Roche) and iron beads. Tubes were then placed in *Qiagen* Retsch MM 300 tissue lyser and shook 4 times for 45 s (frequence 20). Samples were then centrifuged at 12000RPM for 10 min at 4C. Pellets were discarded and supernatants were sonicated (20 s, 30% amplitude). Protein concentration was determined using BCA Protein Assay Kit (Thermo Scientific). Protein extracts were then analyzed by SDS-PAGE as described (20 ug protein extract were loaded per well). Actin was used as a loading control. The rabbit polyclonal antibody KJC8 specific of the beta p53 isoforms was diluted at 1 µg/ml in 4.5% milk in PBS/Tween and incubated overnight at 4C.

## Migration and invasion assays in Boyden chambers

The quantification of cell invasion was carried out in Transwell cell culture chambers containing fluorescence-blocking polycarbonate porous membrane inserts (Fluoroblock; #351152; BD Biosciences; pore size 8 µm). 100 µl of 1 mg/mL Matrigel with reduced growth factors (a commercially prepared reconstituted BM from Englebreth-Holm-Swarm tumors, # 354230; BD Biosciences) were prepared in a Transwell. Cells were transfected as monolayers before trypsinization and plating in 2% FCS-containing media on top of a thick layer (around 500 µm) of Matrigel contained within the upper chamber of a Transwell. Controls were left untreated. The upper and lower chambers were then filled with, respectively, 2% FCS- containing media and media with 10% FCS, thus establishing a soluble gradient of chemo-attractant that permits cell invasion throughout the Matrigel. Cells were allowed to invade at 37°C, 5% $CO_2$ through the gel before fixing for 15 min in 3.7% formaldehyde. Cells that had invaded through the Matrigel were detected on the lower side of the filter by GFP fluorescence and counted. Six fields of the filter were counted and each assay was performed at least three times in triplicate for each condition.

## FACS analysis

Cells were transfected with the GFP-tagged isoforms of p53, harvested 24 hr after transfection and then spun down at 1200 rpm for 5 min and fixed by adding 1 mL of 70% EtOH at −20°C. For each condition, propidium iodide staining was performed and the number of GFP-expressing cells was compared to total cells after quantification by FACS analysis using the CellQuest software.

## Acknowledgement

We are grateful to MRI (Montpellier Rio Imaging) for constructive microscopy, P. Fort for critical comments on the manuscript. The Tayside Tissue Bank is supported by Breast Cancer Campaign, Cancer Research UK (CRUK) and by NHS Tayside through the Chief Scientist Office and Health Sciences Scotland (formerly the Scottish Academic Health Science Collaboration, AHSC). JC Bourdon is a fellow of Breast Cancer Now (2012MaySF127). K Fernandes, J Remenyi, P Quinlan and AM Thompson are supported by Breast Cancer Campaign (BCC: 2010NovPR50, BCC, 2010NovPR16 and BCC, TB2009DUN, respectively). M Khoury was supported by the 'Association pour la Recherche contre le Cancer' (contract AO/3/5099). G Gadea, N Arsic and P Roux are supported by CNRS and INSERM.

## Additional information

### Funding

| Funder | Grant reference number | Author |
|---|---|---|
| Centre National de la Recherche Scientifique | | Gilles Gadea<br>Nikola Arsic<br>Pierre Roux |
| Institut National de la Santé et de la Recherche Médicale | | Gilles Gadea<br>Nikola Arsic<br>Pierre Roux |
| Breast Cancer Campaign | BCC: 2010NovPR50 | Kenneth Fernandes |
| Breast Cancer Campaign | BCC: 2010NovPR16 | Judit Remenyi |
| Association pour la Recherche sur le Cancer | AO/3/5099 | Marie P Khoury |
| Breast Cancer Campaign | BCC: TB2009DUN | Philip R Quinlan<br>Alastair M Thompson |
| Breast Cancer Now | 2012MaySF127 | Jean-Christophe Bourdon |

The funders had no role in study design, data collection and interpretation, or the decision to submit the work for publication.

### Author contributions

GG, NA, KF, AD, SMJ, SA, VM, JR, MPK, PRQ, CAP, LBJ, FVF-P, Acquisition of data, Analysis and interpretation of data, Drafting or revising the article; SV, CA, Acquisition of data, Analysis and interpretation of data, Contributed unpublished essential data or reagents; MdT, MC, Acquisition of data; AMT, Conception and design, Analysis and interpretation of data, Drafting or revising the article; J-CB, PR, Conception and design, Acquisition of data, Analysis and interpretation of data, Drafting or revising the article, Contributed unpublished essential data or reagents

### Author ORCIDs

Frances V Fuller-Pace, http://orcid.org/0000-0001-5859-2932
Jean-Christophe Bourdon, http://orcid.org/0000-0003-4623-9386
Pierre Roux, http://orcid.org/0000-0003-0671-5413

### Ethics

Human subjects: Samples were examined following Local Research Ethics Committee approval under delegated authority by the Tayside Tissue Bank (www.taysidetissuebank.org).

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
