## [Decision Letter]

Thank you for submitting your article "*TP53* drives invasion through expression of its Δ133p53β variant" for consideration by *eLife*. Your article has been favorably evaluated by Charles Sawyers as the Senior Editor and three reviewers, one of whom is a member of our Board of Reviewing Editors.

The reviewers have discussed the reviews with one another and the Reviewing Editor has drafted this decision to help you prepare a revised submission.

The manuscript by Gadea et al. reports a number of interesting observations about the role of a specific mRNA isoform arising from the *TP53* locus, Δ133p53β, in cancer cell invasion and metastasis.

The paper starts with a strong set of observations establishing a correlation between Δ133p53β mRNA expression and poor prognosis in breast cancer. The authors performed a first-in-kind expression analysis for the various Δ133 isoforms and observed that only the β isoform correlates with more aggressive disease and lower survival rates.

Next, the authors embarked on a series of cell-based assays using breast and colorectal carcinoma cell lines and both loss- and gain-of function experiments to demonstrate that Δ133p53β promotes cell motility, invasion and migration, by conferring a sort of 'epithelial to amoeboid' transformation.

Finally, the authors attempt to dissect the mechanism by which Δ133p53β exerts this function, and present some data hinting at a role for the p53 family member p63 as a mediator of Δ133 pro-metastatic functions.

Overall, the reviewers agreed that the work presented is very interesting and the scholarship is very good, but also noted that the last portion of the manuscript (i.e. molecular mechanisms involving p63 isoforms) is poorly developed and the data is not as strong as in the first part of the manuscript.

A main issue is that throughout the first part of the paper the authors demonstrate that Δ133 can exert its functions independently of mutations in DNA binding domain within the *TP53* locus that are commonly observed in cancer, suggesting that its effects are independent of DNA binding. However, during their mechanistic investigations in the later part of the paper, the authors sometimes perform experiments with only the wild type or mutant Δ133 protein, involving different p63 isoforms depending on the cell line employed (ΔNp63 in some scenarios, TAp63 in others). The final product is a rather confusing scenario where it is unclear if mutations in the DBD affect Δ133 function, and also unclear which p63 isoform is involved.

Accordingly, the reviewers agreed to encourage resubmission of a significantly revised manuscript addressing the following Major Points:

1) Removal of the p63 chapter. The reviewers agreed that the paper would be better served by removing the last chapter on mechanism - basically Figures 6 and 7 - and strengthening the main observations throughout Figure 1–Figure 5 by addressing the various technical aspects listed below. Though the association with p63 is of interest, the experiments presented lack major controls and depth to draw solid conclusions. Reviewers concluded that developing the p63 aspect of the paper would require an amount of work not feasible within the review process and that it could become the subject of a follow up study.

2) Improved expression analysis in tumor samples:

2.1) The data on the expression of the different p53 isoforms in tumors only rely on a nested RT-PCR analysis. This could lead to misleading interpretation especially when used to compare expression of different isoforms. For example, are the tumors 5 and 8 in Figure 1 negative or positive for the γ isoform? It is important to verify these data with an orthogonal approach.

2.2) Since an antibody specific for the β isoform is available, it will be important also to verify expression at protein level.

2.3) It is not clear if the analysis in Figure 1—figure supplement 1 was limited to Luminal A tumors. Conducting the analysis without considering breast cancer subtypes could in fact be problematic given the difference p53 isoforms and mutations have different distribution across breast cancer sub-types and given lineage specificities in the behavior of tumors.

3) Another major concern is related to the specificity of the siRNA and their association with the observed phenotypes. A rescue experiment with the appropriate controls will be required to definitely prove that the described differences are due to Δ133 p53 β isoform. Also to have a better sense of the specific functional role of the different p53 isoforms, it will be important that all experiments be performed with all the described siRNAs and upon over expression of the different p53 isoforms.

---

## [Author Response]

*1) Removal of the p63 chapter. The reviewers agreed that the paper would be better served by removing the last chapter on mechanism - basically Figures 6 and 7 - and strengthening the main observations throughout Figure 1–Figure 5 by addressing the various technical aspects listed below. Though the association with p63 is of interest, the experiments presented lack major controls and depth to draw solid conclusions. Reviewers concluded that developing the p63 aspect of the paper would require an amount of work not feasible within the review process and that it could become the subject of a follow up study.*

We agree that the description of the mechanism involving p63 isoforms could be more appropriate in a follow-up of the present study. As suggested by reviewers, we have removed the results related to p63 (Figures 6 and 7) in the Results and Discussion sections of the revised manuscript.

*2) Improved expression analysis in tumor samples:*

*2.1) The data on the expression of the different p53 isoforms in tumors only rely on a nested RT-PCR analysis. This could lead to misleading interpretation especially when used to compare expression of different isoforms. For example, are the tumors 5 and 8 in Figure 1 negative or positive for the γ isoform? It is important to verify these data with an orthogonal approach.*

We agree with the reviewers that it is essential in such study to minimise the number of false positive and false negative data by improving detection threshold and by identifying precisely the different Δ133p53 isoforms. In addition, it is important to take account of the tumor heterogeneity as some p53 isoforms may be only expressed in a small population of tumor cells. It is well-established that only a small percentage of tumor cells can leave the primary tumor site and spread to other organs to form metastasis. Therefore, the methodology to analyse tumor samples should be highly sensitive and specific.

Two methods could be used: PCR that detects specifically each Δ133p53 mRNA variants and/or immunohistochemistry (IHC) of tumor sections using antibodies specific of the different Δ133p53 protein isoforms.

Currently, there is no antibody specific of Δ133p53α or Δ133p53β or Δ133p53γ. Therefore, the detection and identity of each p53 isoforms in tumor samples can only be ascertained by PCR using different sets of primers that specifically amplify Δ133p53α, Δ133p53β or Δ133p53γ mRNA variants respectively. The reviewers’ comment made us realise that we did not explain well enough that the RT-PCR method used in this manuscript was an improved version of the RT-PCR method previously described (Khoury et al., 2013. Detecting and quantifying p53 isoforms at mRNA level in cell lines and tissues. Methods in molecular biology, 962, 1-14). The RT-PCR method used in this manuscript has a higher sensitivity and specificity that enable to detect and accurately identify the expression of Δ133p53α, Δ133p53β and Δ133p53γ mRNA in each tumor sample. It allows thus to compare their respective association with the clinico- pathological markers and/or patient clinical outcome.

We corrected this in the revised manuscript. We wrote a detailed protocol and described the points that increase the specificity and sensitivity of the nested RT-PCR analysis in the “extended experimental procedures”. The primers sequences are now provided in Table 1.

Reviewers suggested to use an orthogonal approach to verify RT-PCR data, notably regarding the expression of Δ133p53γ (For example, are the tumors 5 and 8 in Figure 1 negative or positive for the γ isoform?) We cannot verify expression of the γ p53 protein isoforms by immunoblot or IHC as there is no antibodies specific of the γ isoforms. However, we verified by immunoblot the expression of Δ133p53β protein in some tumor samples (please see below).

*2.2) Since an antibody specific for the β isoform is available, it will be important also to verify expression at protein level.*

The rabbit polyclonal antibody KJC8 was raised and affinity-purified against the peptide TLQDQTSFQKENC corresponding to the C-terminal region of the different β p53 isoforms. It detects all β p53 protein isoforms. Therefore, it is not currently possible to specifically detect Δ133p53β by IHC in tumors.

To address the reviewers’ comments, we determined the expression of p53 isoforms by western- blot. However, this technique is rudimentary as it destroys large amount of a tumor sample preventing its use for on-going and future clinical studies. We were thus authorised to extract protein only from few large breast tumors. The western-blot is included in the new Figure 1—figure supplement 2 and shows that Δ133p53β protein is detected only in the samples expressing Δ133p53β at the mRNA level. Actin was used as a loading control and as an indicator of protein degradation that could have occurred during extraction. Only one band was observed for Actin indicating that no protein degradation occurred during extraction. This immunoblot validates the nested RT-PCR analysis used in this study. Furthermore, it shows for the first time that Δ133p53β protein is expressed in human tumor tissues. This is an important observation as it demonstrates its biological relevance and justifies the characterisation of its biological activities. Thank you for the suggestion, it strengthens the manuscript.

However, in future studies, we would not recommend to verify the nested RT-PCR analysis by immunoblot because it uses large shunk of precious tumor samples. In addition, sensitivity of immunoblot is much lower than PCR analysis; this could lead to misleading interpretation notably for small tumors.

*2.3) It is not clear if the analysis in Figure 1—figure supplement 1 was limited to Luminal A tumors. Conducting the analysis without considering breast cancer subtypes could in fact be problematic given the difference p53 isoforms and mutations have different distribution across breast cancer sub-types and given lineage specificities in the behavior of tumors.*

Figure 1—figure supplement 1 shows by using Kaplan-Meier univariate analyses that several markers (Breast cancer subtypes, *TP53* mutation status, Δ133p53β expression, tumor size, tumor grade, Number of invaded lymph node) are associated with patient clinical outcomes. The analyses were performed on the entire cohort without segregating tumors according to breast cancer subtypes, *TP53* mutation status and p53 isoform expression. It was not limited to the luminal A tumors. We clarified this in the figure legend.

The purpose of Figure 1—figure supplement 1 is to justify the use of the multivariate Cox- regression analysis to identify, among all the markers (Δ133p53β, the breast cancer subtypes (triple negative, Luminal A, Luminal/HER2+, Luminal B, and HER2+), *TP53* mutation status, tumor grade, lymph node metastasis and tumor size (>20mm)), the independent markers significantly associated with the patient’s clinical outcome. The multivariate Cox’s regression analysis led to the identification of Luminal-A as the most significant independent predictor of disease-free survival and overall survival (Table 4).

The Cox’s regression analysis was then reiterated and limited to the luminal-A breast cancer to identify, among all the markers (Δ133p53β, *TP53* mutation, tumor grade, lymph node metastasis and tumor size (>20mm)), the independent markers significantly associated with the clinical outcome of luminal-A patients. Only Δ133p53β was then identified as the most significant independent predictor of cancer progression and death in the luminal A Breast cancer patients (Table 5). Δ133p53α and Δ133p53γ were also tested in the multivariate Cox’s regression analysis but it did not lead to significant association with clinical outcome. Therefore, the statistical analysis was performed in considering all markers when they were included in the Cox’s regression multivariate analysis. This point has been clarified in the revised manuscript.

(Of note, Figure 1—figure supplement 1 shows that Δ133p53β expression is associated with poor disease free survival in WT *TP53* breast cancer patients using Kaplan-Meier univariate analysis. The Kaplan-Meier univariate analysis was also performed for Δ133p53α and Δ133p53γin the WT *TP53* breast cancer patient cohort but no significant association was found.)

*3) Another major concern is related to the specificity of the siRNA and their association with the observed phenotypes. A rescue experiment with the appropriate controls will be required to definitely prove that the described differences are due to δ-133 p53 β isoform. Also to have a better sense of the specific functional role of the different p53 isoforms, it will be important that all experiments be performed with all the described siRNAs and upon over expression of the different p53 isoforms.*

To confirm the specificity of the siRNA used, we performed a rescue experiment to assess whether the difference in invasive phenotype was due to the loss of expression of ∆133p53ß. We analyzed breast cancer MDA-MB-231 D3H2LN cells depleted of all Δ133p53 isoforms (by using the siRNAs si133-1 or si133-2, which specifically target the 5’UTR of Δ133 mRNAs (Aoubala et al., 2011)). The MDA-MB-231 D3H2LN cells express mutant *TP53* gene mutated at codon 280 (R280K). Expression of the si133-1- and si133-2- resistant mutant Δ133p53β-R280K cDNA rescued invasion (Figure 3 and Figure 3—figure supplement 1). In addition, we show that transfection of mutant Δ133p53β-R280K rescued invasion when MDA-MB-231 D3H2LN cells were previously depleted of all β isoforms by siβ (Figure 3). We also verified that expression of si133- resistant WT Δ133p53β cDNA rescued invasion in the LoVo colon cancer cells previously depleted of all Δ133p53 isoforms. The LoVo cells express WT *TP53* gene. These two experiments confirm the specificity of the siRNAs si133-1, si133-2 and siβ. They confirm that the differences in invasion are mainly due to the ∆133p53ß isoform (whether it is WT or mutated).

Another question raised by the reviewers is the specific functional role of the different p53 isoforms. This question is important since the siRNAs si133-1 or si133-2 used in this study deplete all Δ133p53 isoforms, i.e. Δ133p53α, Δ133p53β and Δ133p53γ. In a rescue experiment, we compared the specific contribution of each Δ133p53 isoforms in rescuing the invasive phenotype promoted by the siRNAs si133-1 or si133-2. As shown in the new Figure 3 and Figure 3—figure supplement 1, all three Δ133p53 isoforms rescued invasion in cells depleted of all ∆133p53 variants.

However, Δ133p53β conferred a significantly higher invasive potential compared to Δ133p53α or Δ133p53γ. The results indicate that all Δ133p53 isoforms promote invasion. However Δ133p53β is significantly more potent than Δ133p53α or Δ133p53γ.

Altogether, the experimental data corroborate the clinical data and support the conclusion that the *TP53* gene promotes cancer cell invasion through expression of its Δ133p53β variant.